# Dual Role of WISP1 in maintaining glioma stem cells and tumor-supportive macrophages in glioblastoma

Weiwei Tao[1], Chengwei Chu[1], Wenchao Zhou[1], Zhi Huang[1], Kui Zhai[1], Xiaoguang Fang[1], Qian Huang[1], Aili Zhang[1], Xiuxing Wang[2], Xingjiang Yu[1], Haidong Huang[1], Qiulian Wu[2], Andrew E. Sloan[3,4], Jennifer S. Yu[1,4,5,6], Xiaoxia Li [4,7], George R. Stark[1,4], Jeremy N. Rich [2] & Shideng Bao [1,4,6✉]

The interplay between glioma stem cells (GSCs) and the tumor microenvironment plays crucial roles in promoting malignant growth of glioblastoma (GBM), the most lethal brain tumor. However, the molecular mechanisms underlying this crosstalk are incompletely understood. Here, we show that GSCs secrete the Wnt-induced signaling protein 1 (WISP1) to facilitate a pro-tumor microenvironment by promoting the survival of both GSCs and tumor-associated macrophages (TAMs). WISP1 is preferentially expressed and secreted by GSCs. Silencing WISP1 markedly disrupts GSC maintenance, reduces tumor-supportive TAMs (M2), and potently inhibits GBM growth. WISP1 signals through Integrin α6β1-Akt to maintain GSCs by an autocrine mechanism and M2 TAMs through a paracrine manner. Importantly, inhibition of Wnt/β-catenin-WISP1 signaling by carnosic acid (CA) suppresses GBM tumor growth. Collectively, these data demonstrate that WISP1 plays critical roles in maintaining GSCs and tumor-supportive TAMs in GBM, indicating that targeting Wnt/β-catenin-WISP1 signaling may effectively improve GBM treatment and the patient survival.

[1] Department of Cancer Biology, Lerner Research Institute, Cleveland Clinic, Cleveland, OH 44195, USA. [2] Division of Regenerative Medicine, Department of Medicine, University of California, San Diego, San Diego, CA 92037, USA. [3] Brain Tumor and Neuro-Oncology Center, Seidman Cancer Center, University Hospitals, Case Western Reserve University, Cleveland, OH 44106, USA. [4] Case Comprehensive Cancer Center, Case Western Reserve University School of Medicine, Cleveland, OH 44106, USA. [5] Department of Radiation Oncology, Taussig Cancer Institute, Cleveland Clinic, Cleveland, OH 44195, USA. [6] Center for Cancer Stem Cell Research, Lerner Research Institute, Cleveland Clinic, Cleveland, OH 44195, USA. [7] Department of Inflammation and Immunity, Lerner Research Institute, Cleveland Clinic, Cleveland, OH 44195, USA. ✉email: baos@ccf.org

Glioblastoma (GBM), the WHO grade IV glioma, is the most common and lethal type of primary brain tumor. Despite aggressive treatments, including surgical resection, radiotherapy, and chemotherapy, the median survival of GBM patients remains less than 16 months[1,2]. GBM displays striking cellular heterogeneity and hierarchy, with heterogeneous cancer cells, including glioma stem cells (GSCs) and non-stem tumor cells (NSTCs), in the tumor microenvironment, which also includes endothelial cells, vascular pericytes, abundant tumor-associated macrophages (TAMs), and other immune cells[3,4]. GSCs, comprising a small fraction of cancer cells at the apex of the differentiation hierarchy, play crucial roles in tumor initiation, cancer invasion, tumor angiogenesis, immune evasion, and therapeutic resistance[3,5,6]. GSCs actively interact with other cells in the tumor microenvironment to promote malignant progression in GBMs[4,7,8]. Thus, targeting GSCs and their interactions with other components of the tumor microenvironment has the potential for improving GBM treatment.

TAMs are abundant in the GBM microenvironment, and are important in supporting malignant growth and progression. The density of TAMs correlates positively with glioma grade and negatively with prognosis[9,10]. TAMs are the main source of cytokines that promote tumor cell growth in GBMs, including IL-6[11]. In addition, TAMs closely interact with GSCs[12,13], as both cell types are enriched in perivascular regions and hypoxia niches in GBMs[14–18]. Interestingly, both GSC and TAM populations are increased in recurrent tumors after irradiation[19,20]. Recent studies have shown that TAMs secrete cytokines, such as Pleiotrophin (PTN) and TGF-β1, to promote GSC maintenance and invasion[21,22]. Furthermore, GSCs recruit monocyte-derived TAMs from peripheral blood to promote GBM growth through paracrine Periostin (POSTN) and Osteopontin signaling[18,23]. It is well recognized that TAMs include two major populations: tumor-supportive M2 macrophages and tumor-suppressive M1 macrophages[24], although each population may contain subpopulations. M2 TAMs play immune suppressive roles in the tumor microenvironment to promote tumor growth[25]. The majority of TAMs in GBMs display M2-like properties[26]. These M2 TAMs have been shown to support malignant growth in GBM tumors[18,21,27]. Despite the significant effect of M2 TAMs on GBM progression, the molecular mechanisms underlying the maintenance of M2 TAMs in the tumor microenvironment remain unclear. We have previously shown that GSCs secrete Periostin to recruit monocyte-derived TAMs into GBMs[18], but how TAMs are educated and maintained as M2 tumor-supportive macrophages in the tumor microenvironment in GBM has not been defined.

The Wnt/β-catenin signaling regulates cell proliferation, migration, and death and plays key roles in development, tissue homeostasis, and cancer progression[28]. The activation of the Wnt/β-catenin pathway leads to the stabilization of β-catenin, which is subsequently translocated into the nucleus, where it activates the transcription of Wnt target genes[28]. A previous report showed that tumor-intrinsic Wnt/β-catenin signaling could regulate the tumor microenvironment to promote malignant progression[29]. In lymphoma, Wnt/β-catenin signaling is activated and promotes lymphoma cell chemotaxis towards endothelial cells and adhesion to the endothelial cell layers[30]. Activation of Wnt/β-catenin signaling in melanoma inhibits T cell infiltration to promote tumor growth and therapeutic resistance by regulating CCL4 secretion[31]. In addition, activation of Wnt/β-catenin signaling in osteoblasts promotes metastasis in prostate cancer through paracrine production of WISP1, and activates its receptor on prostate cancer cells in bone metastases[32]. In GBMs, Wnt/β-catenin signaling is highly activated in GSCs, promoting malignant transformation and tumor progression[33,34]. However, how Wnt/β-catenin signaling promotes tumor growth in GBMs is not fully understood. In addition, whether Wnt/β-catenin signaling can regulate the GBM microenvironment to promote malignant progression remains elusive.

To interrogate the potential relationship between Wnt/β-catenin activation and regulation of the tumor microenvironment in GBMs, we analyzed the expression of Wnt/β-catenin-induced secretory proteins, finding that WISP1 is the only highly expressed gene in GBMs relative to normal brains. WISP1, first discovered as a target gene of the Wnt/β-catenin pathway[35], is a secreted cysteine-rich protein that belongs to the CCN family of matri-cellular proteins. It is involved in cell adhesion, survival, proliferation, differentiation, and migration[36]. Increased WISP1 expression is associated with tumor progression in certain tumor types and predicts poor prognosis[37]. A recent study demonstrated that WISP1 is highly expressed in colon cancer and promotes proliferation and invasion[38]. WISP1 is also upregulated in breast cancer to promote cell proliferation, invasion, and epithelial-mesenchymal-transition (EMT)[39]. Here, we investigate the role of WISP1 in regulating GBM growth, finding that WISP1 plays a dual role in promoting GBM growth through both autocrine and paracrine effects. WISP1 promotes GSC maintenance in an autocrine loop. Importantly, it also promotes the survival of tumor-supportive TAMs (M2) to support tumor growth in a paracrine fashion. Inhibition of Wnt/β-catenin-WISP1 signaling by carnosic acid (CA) disrupts the GSC maintenance, inhibits survival of tumor-supportive TAMs, and suppresses GBM growth, suggesting that targeting this signaling axis may effectively improve GBM treatment.

## Results

**WISP1 is preferentially secreted by glioma stem cells.** To investigate the potential molecular link between Wnt/β-catenin signaling and regulation of the tumor microenvironment in GBMs, we analyzed the expression of Wnt/β-catenin target genes, especially secretory proteins, including *CXCL12*, *DKK1*, *WISP1*, *FGF20*, and *EDN1*[40,41], in GBMs, using the TCGA[42] and Gravendeel[43] databases. These analyses revealed that *WISP1* is the only Wnt/β-catenin target gene preferentially expressed in human GBMs relative to normal brain tissues (Fig. 1a, b and Supplementary Fig. 1a, b). Bioinformatic analyses of these databases indicated that high expression of *WISP1* correlates with poor survival (Fig. 1c, d). To assess whether WISP1 is expressed in GBMs, we initially examined WISP1 expression in 5 pairs of matched GSCs and non-stem tumor cells (NSTCs). Matched GSCs and NSTCs were isolated from human GBM surgical specimens or patient-derived GBM xenografts through cell sorting (CD15+/CD133+ for GSCs and CD15−/CD133− for NSTCs). Isolated GSCs were characterized by the expression of the GSC markers (SOX2, OLIG2, CD133, L1CAM) and functional assays including serial neurosphere formation assay, in vitro cell differentiation assay and in vivo limiting dilution tumor formation assay. Immunoblot analyses showed that WISP1, active β-catenin, total β-catenin and the GSC markers including SOX2 and OLIG2 were preferentially expressed in GSCs relative to matched NSTCs (Fig. 1e). Consistently, immunofluorescent staining of WISP1 and the GSC marker SOX2 in matched GSCs and NSTCs validated the preferential expression of WISP1 in GSCs (Fig. 1f). As WISP1 is a secreted protein, we determined the levels of WISP1 in the conditioned media from paired GSCs and NSTCs, confirming that conditioned medium from GSCs contains much more WISP1 than that from matched NSTCs (Fig. 1g). To further verify the preferential expression of WISP1 by GSCs in vivo, we examined the expression patterns of WISP1 in several human

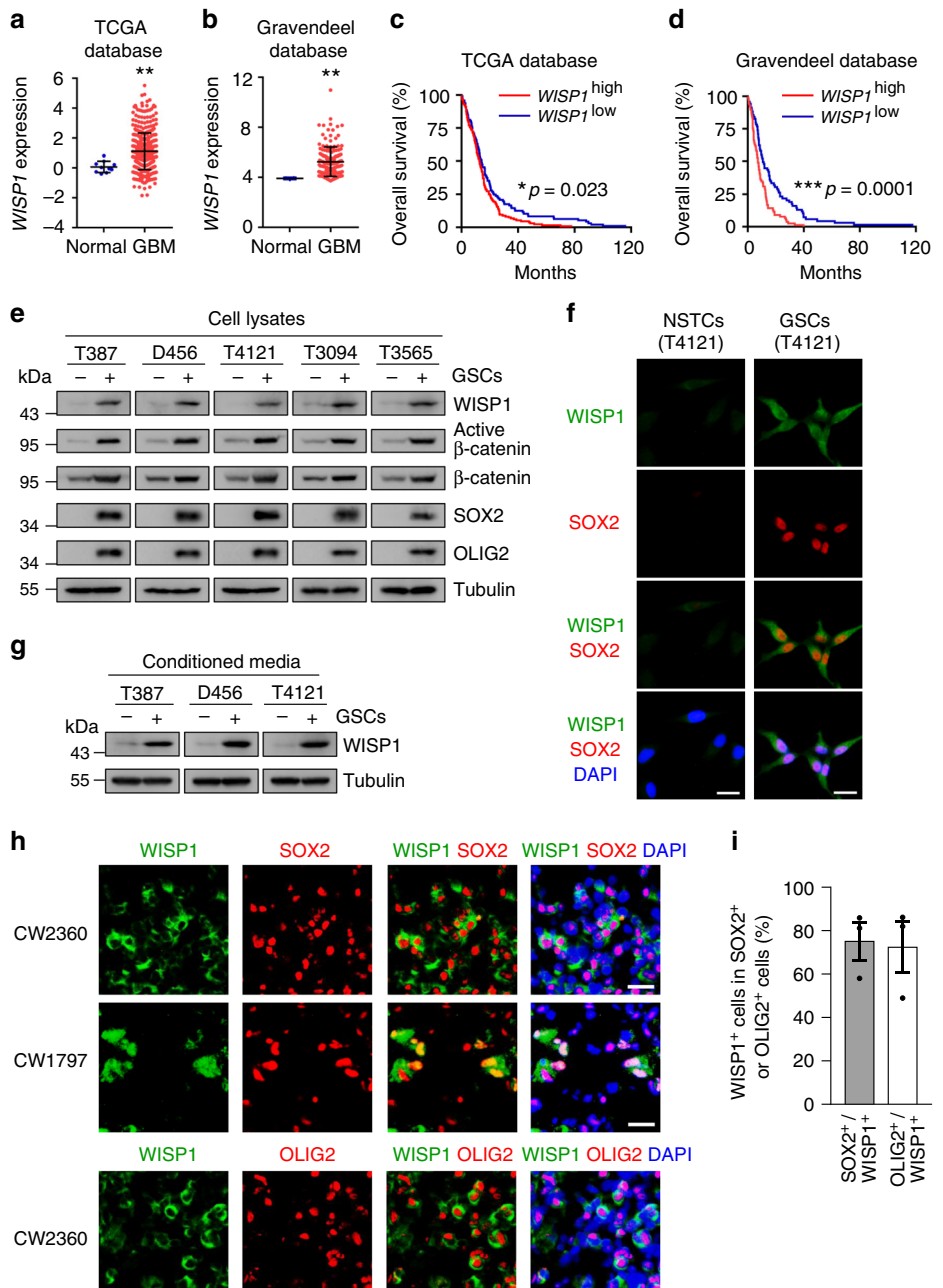

**Fig. 1 WISP1 predicts poor prognosis for GBM patients and is preferentially secreted by GSCs. a** WISP1 expression in human normal brain tissues and GBM tumor samples from the TCGA database. Normal, $n = 10$; GBM, $n = 488$. Data are shown as means ± s.d. **$p = 0.0077$, two-tailed unpaired $t$-test. **b** WISP1 expression in human normal brain tissues and GBM samples from the Gravendeel database. Normal, $n = 8$; GBM, $n = 159$. Data are shown as means ± s.d. **$p = 0.0017$, two-tailed unpaired $t$-test. **c** Kaplan–Meier survival analysis of WISP1 expression and the overall survival of GBM patients from the TCGA database. WISP1 high, $n = 157$; WISP1 low, $n = 122$. *$p = 0.023$, log-rank test. **d** Kaplan–Meier survival analysis of WISP1 expression and the overall survival of GBM patients from the Gravendeel database. WISP1 high, $n = 80$; WISP1 low, $n = 79$. ***$p = 0.0001$, log-rank test. **e** Immunoblot analyses of WISP1, active β-catenin, total β-catenin, SOX2, and OLIG2 expression in cell lysates of GSCs (+) and matched non-stem tumor cells (NSTCs) (–). **f** Immunofluorescent staining of WISP1 (green) and SOX2 (red) in T4121 GSCs and matched NSTCs. Scale Bar, 20 μM. **g** Immunoblot analysis of WISP1 expression in conditioned media from GSCs (+) and matched NSTCs (–). GSCs and matched NSTCs were cultured in neurobasal media without supplements for 40 h and the media was concentrated by vacuum centrifugation. Tubulin in corresponding cell lysates was used as a loading control. **h** Immunofluorescent staining of WISP1 (green) and the GSC marker SOX2 or OLIG2 (red) in human primary GBMs. WISP1 was preferentially expressed in GSCs and distributed in the area near GSCs. Scale Bar, 20 μM. **i** Graphical analysis of (**h**) showing the fraction of WISP1+ cells in SOX2+ or OLIG2+ cells in human primary GBMs. More than 70% of SOX2+ or OLIG2+ GSCs showed WISP1 staining. $n = 3$ independent GBMs. Data are represented as means ± s.e. m. Source data are provided as a Source data file.

GBM specimens and GSC-derived GBM xenografts. Immuno-fluorescent staining confirmed that WISP1 was preferentially expressed in glioma cells expressing the GSC markers SOX2 and OLIG2, and was enriched in the proximity of GSCs (Fig. 1h, i and Supplementary Fig. 1c,d). Taken together, these data demonstrate that WISP1 is preferentially expressed and secreted by GSCs in human GBMs.

**WISP1 supports the maintenance of glioma stem cells**. To determine the functional significance of the preferential expression of WISP1 in GSCs, we examined the effect of WISP1 disruption by shRNA on the GSC maintenance. Silencing WISP1 by two independent shRNAs significantly reduced WISP1 expression in GSCs (Fig. 2a), resulting in decreased GSC proliferation in T4121 and T387 GSCs (Fig. 2b). Furthermore, disruption of WISP1 impaired the self-renewal of GSCs, as assessed by tumorsphere formation (Fig. 2c, d) and in vitro limiting dilution assays (Fig. 2e). These data indicate that WISP1 is required for the GSC maintenance. To further validate the function of WISP1 in GSC maintenance, we also examined whether the exogenous recombinant human WISP1 (rWISP1) protein could replace autocrine WISP1 to rescue GSC proliferation and tumorsphere formation that had been impaired by silencing endogenous WISP1. Consistently, addition of exogenous rWISP1 partially rescued the decreased GSC proliferation and tumorsphere formation caused by WISP1 disruption in a dose-dependent manner (Fig. 2f and Supplementary Fig. 2a). In addition, treatment of GSCs with exogenous rWISP1 increased GSC proliferation in a dose-dependent manner (Supplementary Fig. 2b). Moreover, forced expression of WISP1 in GSCs further augmented cell proliferation and tumorsphere formation (Fig. 2g-i and Supplementary Fig. 2c–e). Collectively, these data indicate that WISP1, secreted by GSCs, promotes cell proliferation and self-renewal of GSCs through an autocrine loop.

**Silencing WISP1 inhibits GSC-driven tumor growth**. As WISP1 plays a critical role in maintaining GSCs, which potentially promotes malignant growth, we examined the impact of disrupting WISP1 on GSC-driven tumor growth in vivo. GSCs (T4121 or T387) expressing firefly luciferase along with WISP1 shRNAs or non-targeting control shRNA (shNT) were transplanted into the brains of immunocompromised mice. Bioluminescent imaging showed that WISP1 disruption markedly impaired GSC-driven tumor growth (Fig. 3a, b and Supplementary Fig. 3a, b). As a consequence, mice bearing xenografts derived from shWISP1-expressing GSCs survived significantly longer than control mice (Fig. 3c and Supplementary Fig. 3c). Immunofluorescent staining indicated that the xenografts from shWISP1-expressing GSCs contained fewer Ki67-postive proliferative cells (Fig. 3d, e and Supplementary Fig. 3d,e) and more apoptotic cells, marked by cleaved-caspase-3 (Fig. 3f, g and Supplementary Fig. 3f, g). In addition, WISP1 disruption by shRNA significantly decreased the GSC population as measured by SOX2 immunofluorescence in tumor xenografts (Fig. 3h, i and Supplementary Fig. 3h,i). Collectively, these results demonstrate that WISP1 plays an essential role in promoting GSC-driven tumor growth in GBMs.

**WISP1 activates the Akt pathway to promote GSC proliferation**. To understand how WISP1 promotes GSC maintenance and tumor growth, we used a protein phospho-kinase array to identify downstream mediators of WISP1 function. The result showed that the activating phosphorylation of Akt (pAkt-Ser473) was dramatically reduced by disrupting WISP1 (Fig. 4a), indicating that WISP1 may regulate Akt activity in GSCs. Immunoblot analysis confirmed that knockdown of WISP1 reduced Akt-

activating phosphorylation (pAkt-Ser473) in GSCs (Fig. 4b), whereas overexpression of WISP1 enhanced this phosphorylation (Fig. 4c). Decreased Akt phosphorylation was also detected in GSC-derived xenografts expressing shWISP1 relative to the shNT control (Supplementary Fig. 4a). As WISP knockdown reduced Akt phosphorylation (pAKT-Ser473), we further examined whether ectopic expression of WISP1 rescues the effect induced by WISP1 disruption. As the shWISP1-2 targets the 3'-end non-coding region of endogenous WISP1 mRNA, and the WISP1 overexpression construct does not contain the 3'-end non-coding sequence, we were able to simultaneously silence endogenous WISP1 and overexpress exogenous WISP1 in GSCs. Immunoblot analyses showed that ectopic expression of WISP1 in GSCs rescued the decreased Akt phosphorylation (pAkt-Ser473) caused by knockdown of endogenous WISP1 (Supplementary Fig. 4b). To further address whether WISP1 is an autocrine agonist of Akt signaling in GSCs, we examined the effect of rWISP1 protein on Akt activation. Consistently, stimulation with rWISP1 induced significant phosphorylation of Akt in GSCs (Fig. 4d). rWISP1 treatment also rescued the decreased Akt phosphorylation (pAkt-Ser473) caused by WISP1 disruption in a dose-dependent manner (Supplementary Fig. 4c). These results demonstrate that WISP1 secreted by GSCs regulates Akt signaling in an autocrine manner. We next explored whether Akt signaling is required for WISP1-mediated GSC maintenance and tumorigenic potential. As Akt1 is the predominant isoform expressed in GSCs, a constitutively active form of Akt1 (Myr-Akt1) was introduced into GSCs expressing shWISP1, or a shNT control (Fig. 4e and Supplementary Fig. 4d). Ectopic expression of Myr-Akt1 restored the proliferation and tumorsphere formation of GSCs impaired by WISP1 disruption (Fig. 4f, g and Supplementary Fig. 4e, f). Similarly, ectopic expression of Myr-Akt1 in GSCs expressing shWISP1 partially restored tumor growth and reduced the survival of mice bearing GSC-derived GBMs (Fig. 4h–j). Immuno-fluorescent staining showed that overexpression of Myr-Akt1 partially rescued the proliferation of GSCs expressing shWISP1 in vivo, as indicated by elevated Ki67-positive staining in xenografts (Supplementary Fig. 4g, h). Collectively, these data suggest that WISP1 activates Akt signaling in GSCs to promote cell proliferation and survival, which may partially augment tumor growth in vivo.

**Integrin α6β1 is a receptor for autocrine WISP1 in GSCs**. To understand the molecular mechanisms underlying WISP1-mediated Akt activation in GSCs, we sought to identify the receptor for the autocrine function of WISP1. Emerging evidence suggests that WISP1 may trigger its downstream signaling by binding to versatile cell surface receptors, the Integrins[44,45]. As a superfamily of cell adhesion receptors, Integrins regulate a variety of cellular responses through various combinations of α and β subunits in a cell-specific manner[46,47]. However, the specific Integrin mediating the function of WISP1 in GSCs was unclear, although Integrins α3, α6, and α7 have been reported to be preferentially expressed in GSCs and promote GSC maintenance and tumor growth[48–50]. To determine specific Integrins that participate in the function of WISP1 in GSCs, we utilized blocking antibodies. Immunoblot analysis showed that anti-Integrin α6 antibody dramatically attenuated Akt activity induced by WISP1 overexpression in GSCs, while the other two blocking antibodies against Integrin α3 or α7 had little effect (Fig. 5a, b). Treatment with the Integrin α6-blocking antibody also reversed the enhanced GSC proliferation and tumorsphere formation induced by ectopic expression of WISP1 (Fig. 5c, d and Supplementary Fig. 5a, b). Because Integrin α6 forms a functional dimer with Integrin β1 in GSCs[49], we used Integrin β1-blocking antibody to

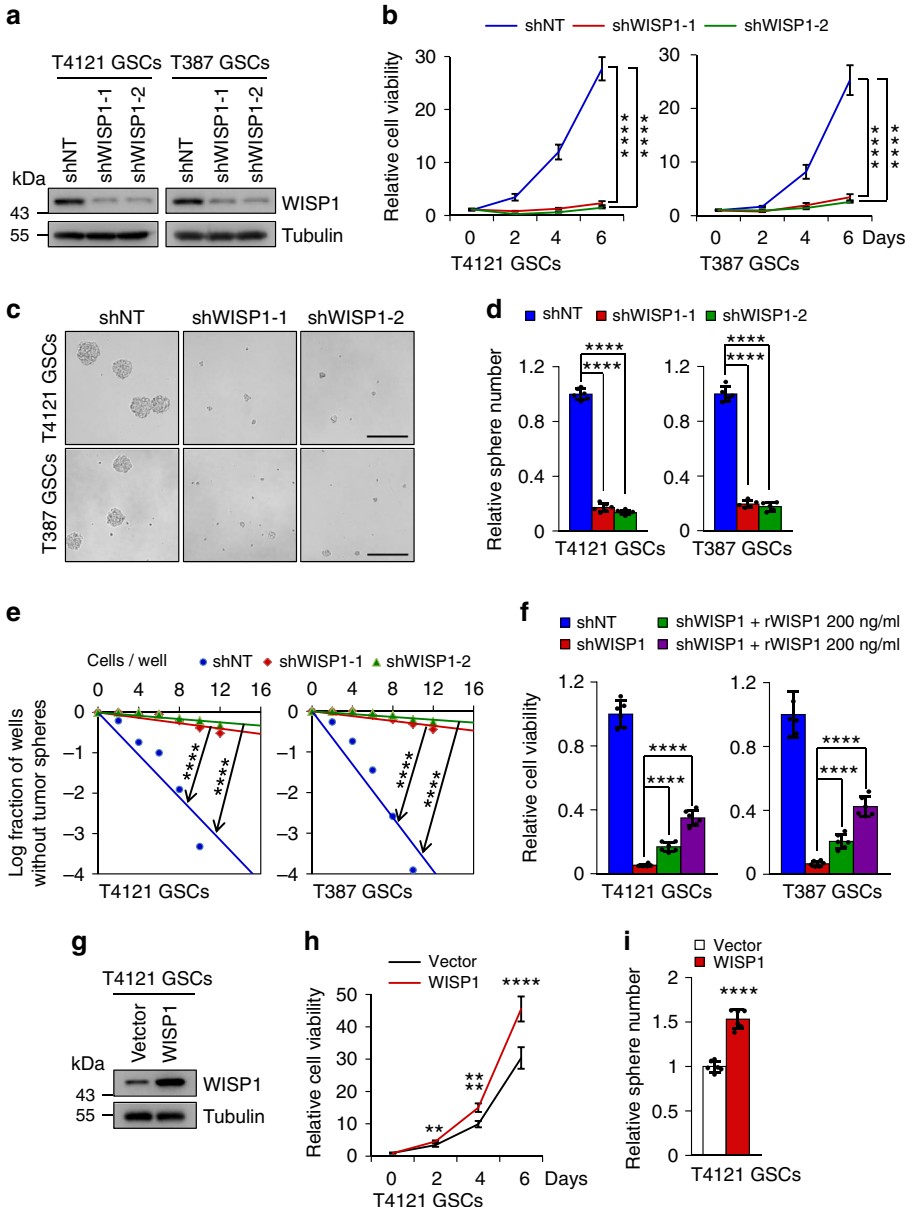

**Fig. 2 WISP1 is required for GSC proliferation and self-renewal. a** Immunoblot analysis of WISP1 expression in GSCs transduced with non-targeting shRNA (shNT) or WISP1 shRNA (shWISP1) through lentiviral infection. **b** Cell viability assay of GSCs transduced with shNT or shWISP1. $n = 6$ (T4121 GSCs) or 5 (T387 GSCs) biological independent samples. Data are shown as means ± s.d. ****$p$ <0.0001, two way ANOVA analysis followed by Tukey's multiple test. **c** Representative tumorsphere images of GSCs transduced with shNT or shWISP1. Scale Bar: 200 μM. **d** Quantification of tumorspheres formed by GSCs transduced with shNT or shWISP1. $n = 6$ (T4121 GSCs) or 5 (T387 GSCs) biological independent cell cultures. Data are shown as means ± s.d. ****$p$ <0.0001, two-tailed unpaired $t$-test. **e** In vitro extreme limiting dilution analysis of the tumorsphere formations of GSCs expressing shNT or shWISP1. $n = 30$ biological independent cell cultures. ****$p$ <0.0001 by ELDA analysis. **f** Cell viability assay of GSCs transduced with shNT or shWISP1 and cultured with different dose of recombinant human WISP1 (rWISP1) protein for 4 days. $n = 6$ biological independent samples. Data are represented as means ± s.d. ****$p$ <0.0001, two-tailed unpaired $t$-test. **g** Immunoblot analysis of WISP1 expression in T4121 GSCs after transduction with WISP1 overexpression or vector control. **h** Cell viability assay of T4121 GSCs transduced with WISP1 overexpression or vector control. $n = 6$ biological independent samples. Data are represented as means ± s.d. **$p$ = 0.0014, ****$p$ <0.0001, two-tailed unpaired $t$-test. **i** Tumorsphere formation of T4121 GSCs transduced with WISP1 overexpression or vector control. $n = 6$ biological independent cell cultures. Data are shown as means ± s.d. ****$p$ <0.0001, two-tailed unpaired $t$-test. Source data are provided as a Source data file.

probe the role of Integrin β1. As expected, anti-Integrin β1 inhibited the GSC proliferation and tumorsphere formation induced by WISP1 overexpression (Fig. 5e, f and Supplementary Fig. 5c, d). Moreover, treatment of GSCs with Integrin α6- or β1-blocking antibody significantly decreased GSC proliferation and tumorsphere formation (Fig. 5g, h and Supplementary Fig. 5e, f). However, blocking Integrin β4, the other binding partner of

Integrin α6, had no effect on GSC proliferation and tumorsphere formation (Fig. 5g, h and Supplementary Fig. 5e,f). Immunoblot analysis confirmed that inhibiting Integrin α6 or β1 by blocking antibody reduced Akt phosphorylation (pAkt-Ser473) in GSCs, while inhibiting Integrin β4 had no effect on the Akt phosphorylation (Supplementary Fig. 5g). We next examined the effects of Integrin α6 disruption by shRNA on GSC proliferation and Akt

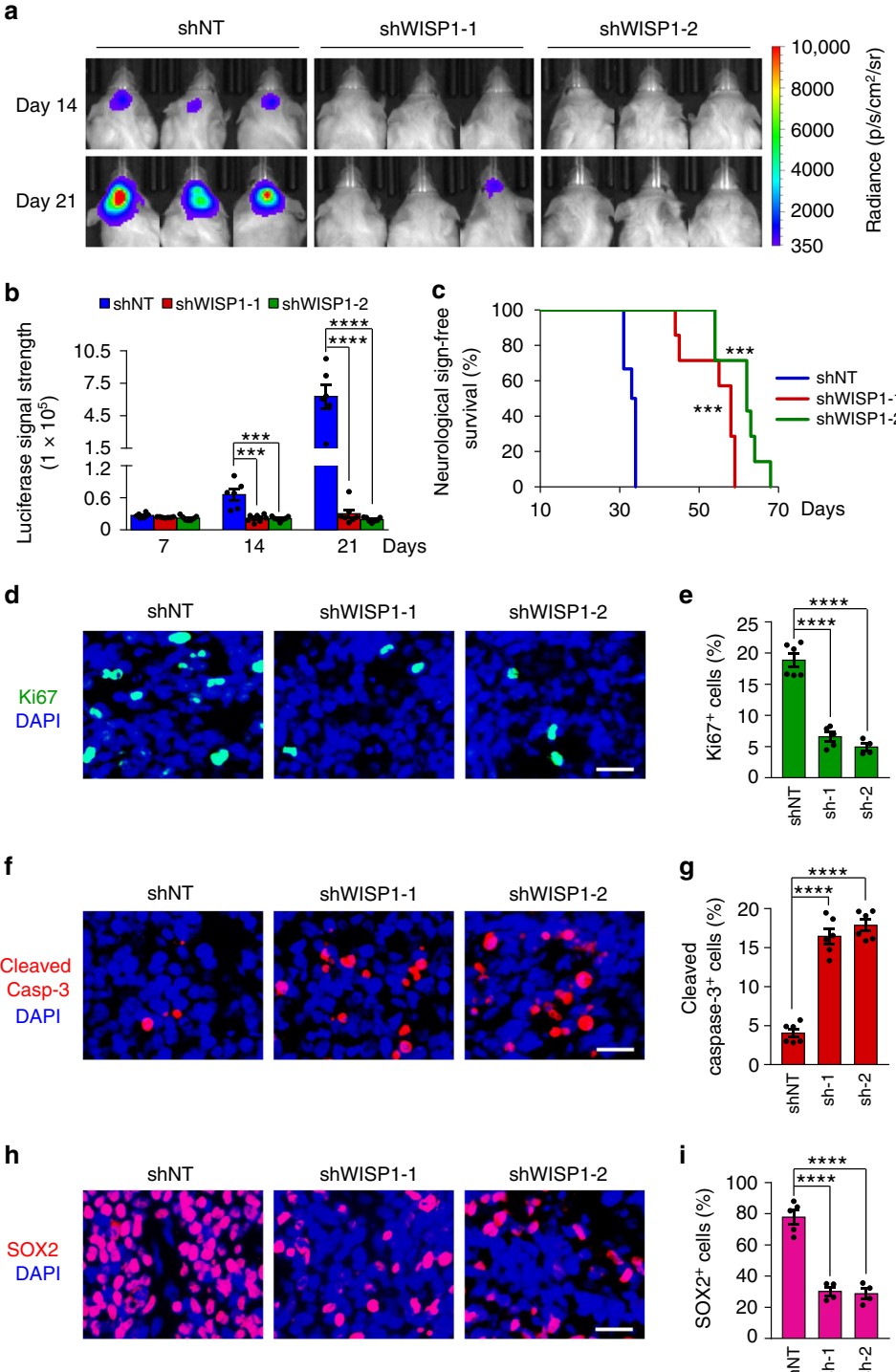

phosphorylation. shRNAs targeting α6 significantly decreased Integrin α6 expression and Akt phosphorylation (pAkt-Ser473) in GSCs (Fig. 5i and Supplementary Fig. 5h). Disruption of Integrin α6 also significantly inhibited GSC proliferation and tumorsphere formation (Fig. 5j, k and Supplementary Fig. 5i, j). Taken together, these data demonstrate that WISP1 enhances Akt activating phosphorylation and GSC proliferation through Integrin α6β1.

To validate that Integrin α6β1 is a receptor for WISP1, we performed co-immunoprecipitation (CoIP) assay to confirm their binding. To increase the potential binding for detection, we overexpressed WISP1 in GSCs and then performed CoIP with anti-Integrin α6 or β1 antibody. Anti-Integrin α6 antibody pulled

down the Integrin α6 along with WISP1 and Integrin β1 (Fig. 6a, b). In addition, the anti-Integrin β1 antibody pulled down the Integrin β1 along with WISP1 and Integrin α6 (Fig. 6c, d). To test the specificity of the interaction between WISP1 and the receptor Integrin α6β1, we examined Akt phosphorylation in GSCs treated with rWISP1 along with Integrin blocking antibody at different ratios. Immunoblot analysis showed that 5 μg/ml of Integrin α6 or β1 blocking antibody dramatically prevented the Akt phosphorylation (pAkt-Ser473) induced by 0.2 μg/ml of rWISP1, while this dose of antibody had a relatively little effect on the Akt phosphorylation (pAkt-Ser473) induced by 0.8 μg/ml of rWISP1 (Fig. 6e). However, 10 μg/ml of Integrin α6 or β1 blocking

**Fig. 3 Disrupting WISP1 in GSCs suppressed GBM tumor growth. a, b,** In vivo bioluminescent images (**a**) and quantification (**b**) of the xenografts derived from luciferase-labeled T4121 GSCs expressing shNT control or shWISP1 on the indicated days after implantation. $n = 6$ (shNT) or 7 (shWISP1-1 or shWISP1-2) mice. Data are represented as means ± s.e.m. ***$p = 0.0008$ (shWISP1-1 versus shNT), ***$p = 0.0006$ (shWISP1-2 versus shNT), ****$p$ <0.0001, two-tailed unpaired $t$-test. p, photons; sr, steradian. **c** Kaplan–Meier survival curves of mice implanted with T4121 GSCs expressing shNT or shWISP1. $n = 6$ (shNT) or 7 (shWISP1-1 or shWISP1-2) mice. ***$p = 0.0004$, log-rank test. **d, e** Immunofluorescent staining of Ki67 (green, **d**) and quantification of Ki67$^+$ cells (**e**) in GBM tumors derived from T4121 GSCs expressing shNT, shWISP1-1 (sh1) or shWISP1-2 (sh-2). $n = 6$ (shNT) or 5 (shWISP1-1) or 4 (shWISP1-2) biological independent tumor tissues. Scale bar: 25 μm. Data are shown as means ± s.e.m. ****$p$ <0.0001, two-tailed unpaired $t$-test. **f, g,** Immunofluorescent staining of Cleaved Caspase-3 (Cleaved Casp-3, Red, **f**) and quantification of Cleaved Caspase-3$^+$ cells (**g**) in GBM tumors derived from T4121 GSCs expressing shNT, shWISP1-1 or shWISP1-2. $n = 6$ biological independent tumor tissues. Scale bar: 25 μm. Data are represented as means ± s.e.m. ****$p$ <0.0001, two-tailed unpaired $t$-test. **h, i** Immunofluorescent staining of SOX2 (Red, **h**) and quantification of SOX2$^+$ cells (**i**) in GBM tumors derived from T4121 GSCs expressing shNT, shWISP1-1 or shWISP1-2. $n = 5$ (shNT) or 4 (shWISP1-1 or shWISP1-2) biological independent tumor tissues. Scale bar: 25 μm. Data are shown as means ± s.e.m. ****$p$ <0.0001, two-tailed unpaired $t$-test. Source data are provided as a Source data file.

antibody dramatically prevented the increased Akt phosphorylation (pAkt-Ser473) induced by both doses of rWISP1 (Fig. 6e). These results indicate that Integrin α6β1 is relatively specific to WISP1. Moreover, immunofluorescent analysis showed the co-expression of WISP1 and Integrin α6 proteins in primary human GBM samples (Fig. 6f). A recent study showed that malignant cells in human GBM exist in four main cellular states that recapitulate neural-progenitor-like (NPC-like), oligodendrocyte-progenitor-like (OPC-like), astrocyte-like (AC-like), and mesenchymal-like (MES-like) states[51]. Thus, we also assessed the expression of WISP1 and Integrin α6β1 across the four GBM cellular states. The results showed that WISP1 is enriched in some AC-like and MES-like cells, while Integrin α6 and β1 are widely expressed in all four states. These data suggest that WISP1 and Integrin α6β1 are co-expressed by some AC-like and MES-like cells in GBM (Supplementary Fig. 5k). Collectively, these data indicate that Integrin α6β1 is the receptor for autocrine signaling in response to WISP1 in GSCs.

**WISP1 promotes the survival of tumor-supportive TAMs in vivo.** Because overexpression of Myr-Akt1 in GSCs expressing shWISP partially rescued the impaired tumor growth caused by WISP1 disruption, we speculated that other mechanisms may be involved in the growth of GBMs promoted by WISP1, in addition to its autocrine signaling in GSCs. Therefore, we explored whether secreted WISP1 could also affect other cell types in GBMs, in a paracrine manner. First, we examined the impact of rWISP1 on the viability of NSTCs in vitro. In a cell titer assay, exogenous rWISP1 treatment had no obvious effect on the growth or survival of NSTCs (Supplementary Fig. 2f). We also analyzed whether WISP1 disruption could impact tumor angiogenesis. Immunofluorescent staining using the endothelial marker Glut1 showed that WISP1 knockdown had little effect on vascular density in GSC-derived tumors (Supplementary Fig. 6a–d). Because GBMs usually contain abundant TAMs that mainly promote malignant progression[12,52], we examined whether WISP1 disruption could affect TAM density and subtype distribution in GBMs. Immunofluorescent staining using the total TAM markers Iba1 and CD11b demonstrated that knockdown of WISP1 markedly decreased TAM density in GSC-derived xenografts (Fig. 7a–f and Supplementary Fig. 6c, e), and ectopic expression of Myr-Akt1 did not rescue the decreased TAM density caused by WISP1 disruption in the GSC-derived xenografts (Supplementary Fig. 6f, g). The expression of WISP1 were indeed significantly decreased in xenografts expressing WISP1 shRNA, demonstrated that these tumors were not derived from the GSCs that escaped form shRNA knockdown (Fig. 7a–f and Supplementary Fig. 6f, g). As both GSCs and TAMs are enriched in perivascular niches in GBMs, we next examined the potential correlation between WISP1 expression and TAM density in primary GBM specimens. Immunofluorescent analysis showed that TAMs are enriched in the WISP1-abundant regions (Supplementary Fig. 7a), supporting the idea that WISP1 may play a role in the TAM maintenance. As TAMs include both tumor-supportive macrophages (M2 TAMs) and tumor-suppressive macrophages (M1 TAMs)[53,54], we investigated which subtype of TAMs is affected by WISP1 disruption in GSC-derived xenografts. We used several specific M2 markers (CD206, CD163, Arg1, and Fizz1) and M1 markers (CD11c, CD16/32, iNOS, and MHCII) for the study, as those markers have been used to distinguish M2/M1 TAMs in GBM xenograft models from our group[18] and others[55–57]. We found that WISP1 disruption markedly reduced M2 TAMs in GSC-derived tumors (Fig. 7g–l and Supplementary Fig. 7b–g). Interestingly, disrupting WISP1 had little effect on M1 TAMs (Supplementary Fig. 8a–l). Immunofluorescent analyses further demonstrated that disrupting WISP1 increased apoptosis of M2 TAMs (Cleaved Caspase-3$^+$/CD206$^+$) (Supplementary Fig. 9a, b) and showed no effect on M1 TAMs (Cleaved Caspase-3$^+$/CD16/32$^+$) (Supplementary Fig. 9c, d). Consistently, WISP1 disruption resulted in a significant increase in total apoptotic TAMs (Cleaved Caspase-3$^+$/Iba1$^+$) in the xenografts (Supplementary Fig. 9e, f). Taken together, these data demonstrate that WISP1 secreted by GSCs potently promotes the survival of M2 TAMs in GBMs.

To further confirm the WISP1 function in M2 TAM maintenance and GBM tumor growth, we applied a Tet-On inducible expression system to examine whether inducible overexpression of WISP1 in response to doxycycline (Dox) affects the TAM population and GBM tumor growth. Immunoblot analysis confirmed that Dox treatment enhanced WISP1 expression in GSCs (Supplementary Fig. 10a). Bioluminescent imaging demonstrated that induced overexpression of WISP1 by Dox treatment significantly promoted GSC-driven tumor growth in mouse brains (Supplementary Fig. 10b–d). Importantly, induced expression of WISP1 by Dox also increased the density of M2 and total TAMs (Supplementary Fig. 10e–h). These results validate that WISP1 plays a critical role in maintaining M2 TAMs to support GBM tumor growth.

**WISP1 signals via α6β1-Akt to promote M2 TAM survival.** As our data suggest that Integrin α6β1 is the key receptor for WISP1-mediated signaling in GSCs, we next examined whether Integrin α6β1 is also expressed on cell surface of TAMs. Immunofluorescent staining demonstrated that Integrin α6 or β1 was also expressed by tumor-supportive M2 TAMs (Supplementary Fig. 11a, b) but rarely colocalized with the M1 TAM markers in human GBMs (Supplementary Fig. 11c, d). To confirm the preferential expression of α6β1 in M2 TAMs, we further examined α6β1 expression in primed M1 and M2

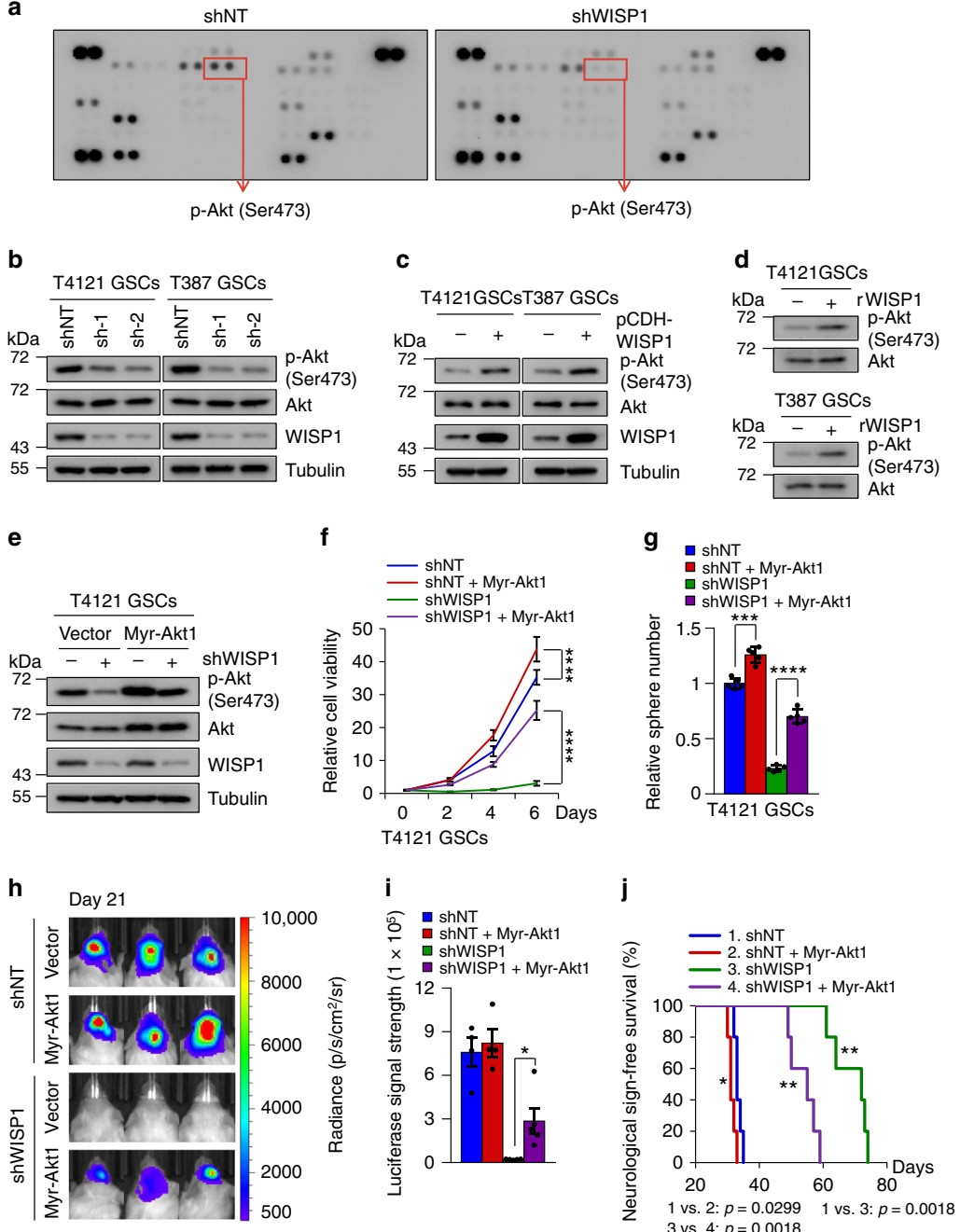

**Fig. 4 WISP1 activates Akt signaling in GSCs. a** Phospho-kinase array of cell lysates from T387 GSCs transduced with shNT or shWISP1. T387 GSCs were transduced with NT or WISP1 shRNA for 48 h and then lysed in RIPA lysis buffer. Lystes were centrifuged and then analyzed by using a phospho-kinase array kit according to the manufacturer's protocol. **b** Immunoblot analysis of Akt activating phosphorylation (Ser473) in GSCs (T4121 and T387) transduced with shWISP1 or shNT control. **c** Immunoblot analysis of Akt activating phosphorylation (Ser473) in GSCs transduced with WISP1 overexpression or vector control. **d** Immunoblot analysis of Akt activating phosphorylation (Ser473) in GSCs in response to rWISP1 protein stimulation. Cells were cultured in neurobasal media without supplements overnight and then treated with rWISP1 (400 ng/ml) for 6 h. **e** Immunoblot analysis of Akt activating phosphorylation (ser473) and WISP1 expression in T4121 GSCs transduced with Myr-Akt1 or vector control in combination with shNT or shWIPS1. **f, g** Cell titler assay (**f**) of T4121 GSCs transduced with Myr-Akt1 or vector control in combination with shNT or shWIPS1 (n = 5 biological independent samples). Relative tumorsphere number is shown (**g**) (n = 5 biological independent cell cultures). Data are shown as means ± s.d. ****p <0.0001, two way ANOVA analysis followed by Tukey's multiple test (**f**). ***p = 0.0001, ****p <0.0001, two-tailed unpaired t-test (**g**). **h, i** Bioluminescent images (**h**) and quantification (**i**) of GBM xenografts derived from luciferase-labeled T4121 GSCs transduced with Myr-Akt1 or vector control in combination with shNT or shWIPS1. n = 4 (shNT or shNT+Myr-Akt1) or 5 (shWISP1 or shWISP1+Myr-Akt1) mice. Representative images and quantification on day 21 posttransplantation are shown. Data are shown as means ± s.e.m. *p = 0.0173, two-tailed unpaired t-test. p, photons; sr, steradian. **j** Kaplan–Meier survival curves of mice implanted with T4121 GSCs transduced with Myr-Akt1 or vector control in combination with shWIPS1 or shNT control. n = 5 mice. *p = 0.0299, **p = 0.0018, log-rank test. Source data are provided as a Source data file.

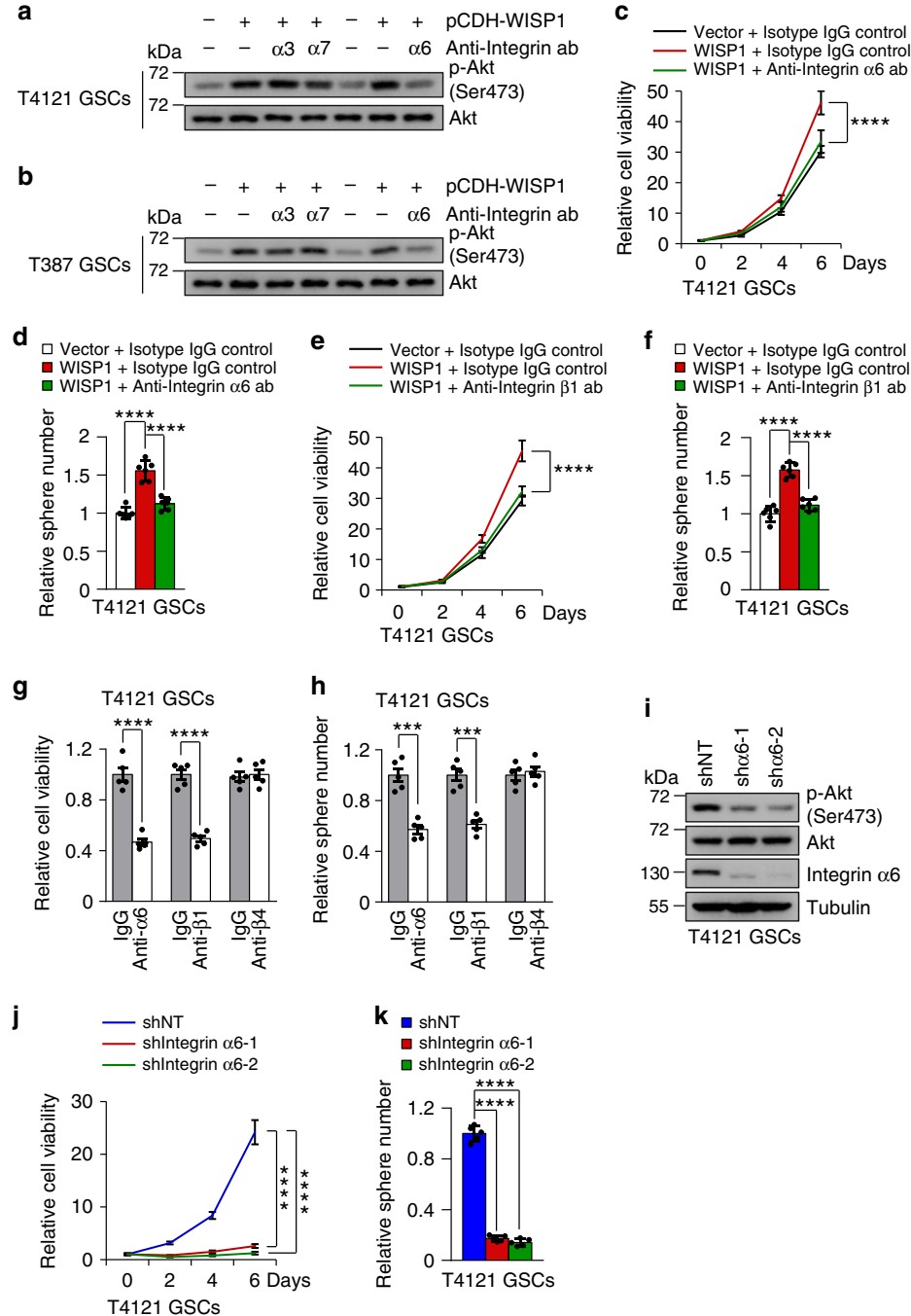

**Fig. 5 WISP1 acts through Integrin α6β1 to activate Akt in GSCs. a**, **b**, Immunoblot analysis of Akt activating phosphorylation (Ser473) in T4121 GSCs (**a**) and T387 GSCs (**b**) treated with 5 μg/ml Integrin blocking antibody (Anti-Integrin ab) or relative isotype IgG control in combination with WISP1 overexpression or vector control for 48 h. α3, Integrin α3; α6, Integrin α6; α7, Integrin α7. **c**, **d**, Cell viability (**c**) or tumorsphere formation (**d**) assay of T4121 GSCs treated with 5 μg/ml Integrin α6 blocking antibody (ab) or isotype IgG in combination with WISP1 overexpression or vector control. $n = 6$ biological independent samples. Data are shown as means ± s.d. ****$p < 0.0001$, two way ANOVA analysis followed by Tukey's multiple test (**c**). ****$p < 0.0001$, two-tailed unpaired $t$-test (**d**). **e**, **f**, Cell viability (**e**) or tumorsphere formation (**f**) assay of T4121 GSCs treated with Integrin β1 blocking antibody (5 μg/ml) or isotype IgG in combination with WISP1 overexpression or vector control. $n = 5$ (**e**) or 6 (**f**) biological independent samples. Data are represented as mean ± s.d. ****$p < 0.0001$, two way ANOVA analysis followed by Tukey's multiple test (**e**). ****$p < 0.0001$, two-tailed unpaired $t$-test (**f**). **g**, **h**, Cell viability (**g**) or tumorsphere formation (**h**) assay of T4121 GSCs treated with Integrin blocking antibody (5 μg/ml) or isotype IgG for 6 days. $n = 5$ biological independent samples. Data are represented as mean ± s.d. ***$p = 0.001$, ****$p < 0.0001$, two-tailed unpaired $t$-test. α6, Integrin α6; β1, Integrin β1; β4, Integrin β4. **i** Immunoblot analysis of Akt phosphorylation (Ser473) and Integrin α6 expression in T4121 GSCs transduced with shIntegrin α6 or shNT control. **j**, **k**, Cell viability (**j**) or tumorsphere formation (**k**) assay of T4121 GSCs transduced with shIntegrin α6 or shNT. $n = 5$ biological independent samples. Data are shown as means ± s.d. ****$p < 0.0001$, two way ANOVA analysis followed by Tukey's multiple test (**j**). ****$p < 0.0001$, two-tailed unpaired $t$-test (**k**). Source data are provided as a Source data file.

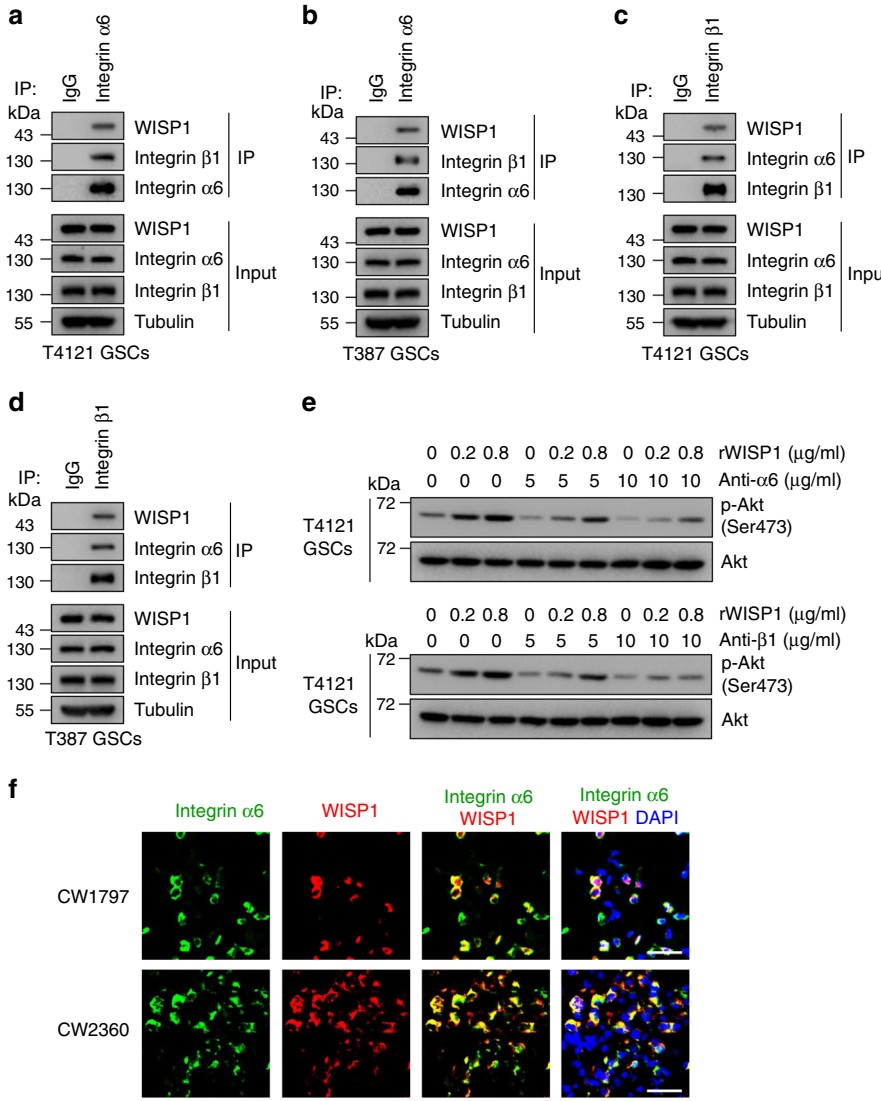

**Fig. 6 WISP1 binds to Integrin α6β1 in GSCs. a**, **b**, CoIP assays of protein interaction in T4121 GSCs (**a**) or T387 GSCs (**b**) transduced with WISP1 overexpression lentivirus. Cell lysates were immunoprecipitated (IP) with anti-Integrin α6 antibody and then immunoblotted with anti-WISP1, anti-Integrin α6 and anti-Integrin β1 antibodies. **c**, **d**, CoIP assays of protein interaction in T4121 GSCs (**c**) or T387 GSCs (**d**) transduced with WISP1 overexpression lentivirus. Cell lysates were immunoprecipitated (IP) with anti-Integrin β1 antibody and then immunoblotted with anti-WISP1, anti-Integrin α6 and anti-Integrin β1 antibodies. **e** Immunoblot analysis of Akt phosphorylation (Ser473) in T4121 GSCs treated with 5 or 10 μg/ml Integrin blocking antibody in combination with 0.2 or 0.8 μg/ml rWISP1 protein for 12 h. **f** Immunofluorescent staining of Integrin α6 (green) and WISP1 (red) in human primary GBM samples. Scale bar, 30 μM.

macrophages in vitro. Since U937 monocyte-like cells can be primed to differentiate into macrophages[18,21], we polarized U937 cells into M1 or M2 macrophages to mimic TAMs for our in vitro study. Immunoblot analyses of the M2 markers (CD163, CD206, and Arg-1) or M1 markers (MHC II and iNOS) validated that U937 cells were successfully polarized into M1 or M2 macrophages (Supplementary Fig. 12a). Consistently, both Integrins α6 and β1 were preferentially expressed in M2 macrophages relative to M1 macrophages (Supplementary Fig. 12a). To verify the function of WISP1 in macrophage survival, we examined whether exogenous rWISP1 protein could rescue the macrophages from serum starvation-induced cell death. Indeed, rWISP1 treatment significantly prevented the death of M2 macrophages in a dose-dependent manner, while having no effect on M1 macrophages (Supplementary Fig. 12b). In addition, rWISP1 treatment enhanced Akt-

activating phosphorylation (pAkt-Ser473) in M2 macrophages, but not in M1 macrophages (Supplementary Fig. 12c). To assess whether WISP1 promotes the survival of M2 TAMs through Integrin α6β1 signaling, we applied Integrin α6 shRNAs to knockdown its expression (Supplementary Fig. 12d). Disruption of Integrin α6 by shRNAs inhibited the rWISP1-enhanced survival of M2 macrophages cultured under serum starvation condition (Supplementary Fig. 12e). To further validate this result, we then applied Integrin α6- or β1-neutralizing antibodies to block this function. Anti-Integrin α6 or anti-β1 also substantially inhibited the rWISP1-enhanced survival of M2 macrophages cultured under serum starvation condition (Supplementary Fig. 12f). Consistently, Integrin α6 or β1 blocking antibody attenuated rWISP1-induced Akt-activating phosphorylation in M2 macrophages (Supplementary Fig. 12g). Collectively, these results indicate that WISP1

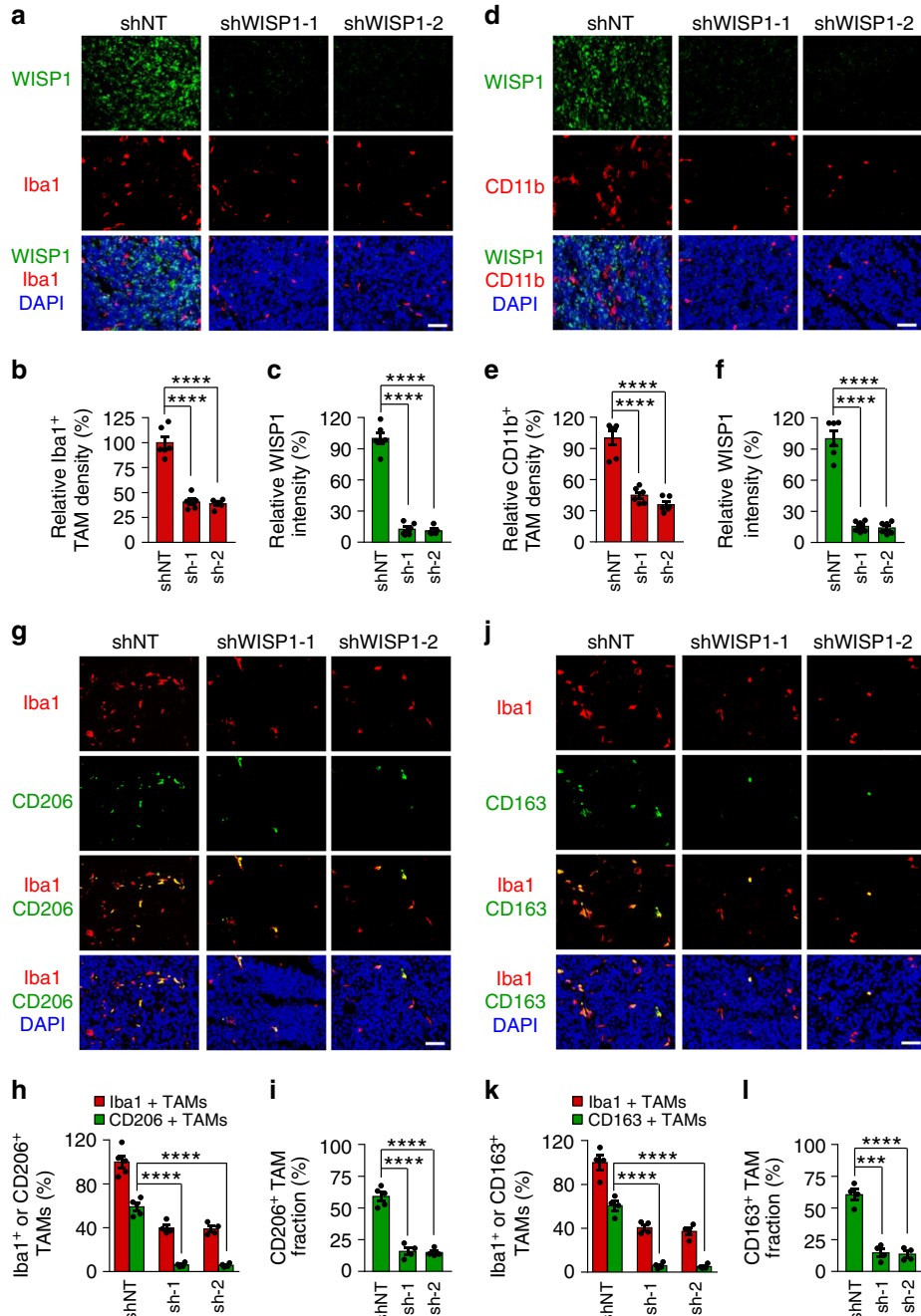

**Fig. 7 Disrupting WISP1 in GSCs reduced TAM density in GBM tumors. a** Immunofluorescent staining of WISP1 (green) and the TAM marker Iba1 (red) in GBM xenografts derived from T4121 GSCs expressing shWISP1 or shNT control. Scale Bar, 50 μM. **b, c** Quantitation of Iba1+ TAM density (**b**) or WISP1 intensity (**c**) in xenografts derived from T4121 GSCs expressing shWISP1 or shNT. $n = 6$ (shNT or shWISP1-1) or 5 (shWISP1-2) biological independent tumor tissues. Data are shown as means ± s.e.m. ****$p$ <0.0001, two-tailed unpaired $t$-test. **d** Immunofluorescent staining of WISP1 (green) and the TAM marker CD11b (red) in xenografts derived from T4121 GSCs expressing shNT or shWISP1. Scale Bar, 50 μM. **e, f** Quantitation of CD11b+ TAM density (**e**) or WISP1 intensity (**f**) in xenografts derived from T4121 GSCs expressing shNT or shWISP1. $n = 6$ biological independent tumor tissues. Data are shown as means ± s.e.m. ****$p$ <0.0001, two-tailed unpaired $t$-test. **g** Immunofluorescent staining of the M2 TAM Marker CD206 (green) and the pan-macrophage marker Iba1 (red) in GBM xenografts derived from T4121 GSCs expressing shNT control or shWISP1. Scale Bar, 50 μM. **h, i** Quantitation of CD206+ TAM (M2) density (**h**) and the fraction of M2 TAMs (**i**) in xenografts derived from T4121 GSCs expressing shNT or shWISP1. $n = 5$ (shNT) or 4 (shWISP1-1 or shWISP1-2) biological independent tumor samples. The M2 TAM fraction was determined by the percentage of M2 TAMs within TAMs in shNT or shWISP1 xenografts, respectively. Data are represented as means ± s.e.m. ****$p$ <0.0001, two-tailed unpaired $t$-test. **j** Immunofluorescent staining of the M2 TAM Marker CD163 (green) and the pan-macrophage marker Iba1 (red) in xenografts derived from T4121 GSCs expressing shNT or shWISP1. Scale Bar, 50 μM. **k, l** Quantitation of CD163+ TAM (M2) density (**k**) and the fraction of M2 TAMs (**l**) in xenografts derived from T4121 GSCs expressing shNT or shWISP1. $n = 4$ biological independent tumor samples. Data are represented as means ± s.e.m. ***$p = 0.001$, ****$p$ <0.0001, two-tailed unpaired $t$-test. Xenografts were collected from mice when neurological signs occur after GSC transplantation. Source data are provided as a Source data file.

promotes the survival of tumor-supportive M2 macrophages by activating Integrin α6β1-Akt signaling.

**Disrupting the Wnt/β-catenin-WISP1 axis inhibits GBM growth.** To evaluate the therapeutic potential of targeting Wnt/β-catenin-WISP1 signaling in GBM, we examined whether pharmacologic inhibition of this pathway by carnosic acid (CA), a small molecule inhibitor of β-catenin activity[58], could impact GSCs and M2 TAMs to inhibit GBM tumor growth. We selected carnosic acid in our preclinical study, because it can penetrate the blood-brain barrier[59,60], and it has been reported to improve the treatment of medulloblastoma in a mouse model[60]. When GSCs were treated with different doses of CA, the expression levels of active β-catenin and WISP1 were significantly reduced in a dose-dependent manner (Supplementary Fig. 13a, b). Consistently, CA treatment markedly reduced GSC viability (Supplementary Fig. 13c, d) and suppressed GSC tumorsphere formation (Supplementary Fig. 13e, f) in a dose-dependent manner. Next, we examined the effect of CA on the growth of orthotopic GBM xenografts, based on its in vitro efficacy and known ability to penetrate the blood-brain barrier[59,60]. In vivo bioluminescent imaging indicated that CA significantly inhibited the growth of GSC-derived xenografts (Fig. 8a–c). Consequently, mice treated with CA had a significantly extended survival relative to the control group (Fig. 8d). Immunofluorescent staining showed that CA administration reduced Ki67-postive proliferative cells (Fig. 8e, f) and increased the number of apoptotic cells, marked by cleaved-caspase-3, in GSC-derived xenografts (Fig. 8g, h). In addition, CA treatment significantly reduced the GSC population in GBM xenografts, as demonstrated by SOX2 immunofluorescence (Fig. 8i, j). Moreover, CA treatment resulted in a significant decrease in WISP1 expression, in the number of M2 TAMs (CD206+ or CD163+) and total TAMs (Iba1+) in GSC-derived xenografts (Fig. 8k, l and Supplementary Fig. 13g–i). Collectively, these data demonstrate that inhibition of Wnt/β-catenin-WISP1 signaling by CA disrupts GSC maintenance, impairs M2 TAM survival, and potently suppresses GBM tumor growth, indicating that targeting this pathway may effectively improve GBM treatment.

## Discussion

Wnt/β-catenin signaling has been implicated in the regulation of malignant growth in several cancer types, but less is known regarding its role in mediating crosstalk between GSCs and other cells in the tumor microenvironment. In this study, we identified WISP1 as a key mediator of the GSC-GSC and GSC-TAM crosstalk in GBMs (Fig. 9). We demonstrate that WISP1 plays crucial roles in promoting the maintenance of GSCs and survival of tumor-supportive M2 TAMs by activating Akt (Fig. 9). Moreover, Inhibition of Wnt/β-catenin-WISP1 signaling markedly suppresses GBM tumor growth, suggesting that targeting this signaling axis represents an attractive therapeutic strategy.

Our findings indicate that WISP1 promotes GSC proliferation and self-renewal in an autocrine loop. Several studies reported that autocrine WISP1 signaling enhanced cell growth in various cancers such as breast cancer and oral squamous cell carcinoma[61–63], but its autocrine role in GBM has not been defined. A recent study showed that WISP1 is an oncogene in GBM and inhibition of WISP1 suppressed the proliferation of GBM cells[64]. However, the origin of WISP1 in GBM and the role of WISP1 in regulating of GSC properties remain unclear. Our study reveals that WISP1 is preferentially expressed by GSCs and activates Akt signaling to promote GSC proliferation. As re-activating Akt signaling in shWISP-expressing GSCs only partially rescues tumor growth, we further investigated additional mechanisms

that might be involved in WISP1-promoted GBM tumor growth and surprisingly found that the WISP1-mediated activation of Akt is crucial for maintaining tumor-supportive M2 TAMs.

TAMs are critical immune cells in the GBM microenvironment and play important roles in facilitating GBM growth[65]. Our study reveals a paracrine mechanism that drives the survival of tumor-supportive M2 TAMs in which the WISP1 produced by GSCs supports GBM malignant progression. Our results demonstrate that disruption of WISP1 dramatically reduces density of M2 TAMs. We fully recognized that the M1/M2 dichotomy is an oversimplification of TAMs in tumors. In this study, we used the term "M2 TAMs" to indicate the tumor-supportive macrophages that may contain several subpopulations, and used "M1 TAMs" to represent the tumor-suppressive macrophages that may also contain subpopulations. The M1/M2 dichotomy used here does not mean that there are only two simple types of TAMs in GBM tumors. We believe that there is a heterogeneity of TAMs in GBM tumors. However, our studies confirmed that silencing WISP1 indeed reduced tumor-supportive macrophages (M2 TAMs) in our xenograft models. According to our previous studies[18,21] and current data, it is reasonable to conclude that M2/M1 TAMs indeed represent two major but functionally different macrophage populations (tumor-supportive and tumor-suppressive) in our tumor models, although we can't rule out that each major population (M2 or M1) may contain several subpopulations. It will be interesting to further analyze subpopulations in M2 TAMs and M1 TAMs in GBMs in the future. A preclinical study reported that blocking CSF-1R (macrophage colony-stimulating factor 1 receptor) did not impact total TAM density in GBMs, indicating that other survival factors from the tumor microenvironment may provide compensatory growth and survival signals[27]. However, we found that silencing WISP1 in GSCs markedly decreased TAM density in GSC-derived GBM tumors. It is possible that silencing WISP1 may result in an altered tumor microenvironment, which may contribute to decreased TAMs. However, our in vitro data suggest that WISP1 has a direct effect on the survival of macrophages. It would be interesting to further investigate whether WISP1 can regulate the tumor microenvironment in GBMs in the future. Our previous study demonstrated that GSCs secrete Periostin to recruit monocyte-derived TAMs into GBMs[18], but how these TAMs are maintained as M2 TAMs in GBM was not clear. In this study, we discover that WISP1, secreted by GSCs, promotes the survival of M2 TAMs and thus maintains tumor-supportive macrophages in GBM tumors, indicating that the recruitment of TAMs and the maintenance of M2 TAMs are regulated by different molecules secreted by GSCs in the tumor microenvironment. Therefore, GSCs play vital roles not only in TAM recruitment but also in the maintenance of M2 TAMs, indicating that GSCs could manipulate their niches through multiple paracrine functions. Similarly, tumor-supportive M2 TAMs may secrete several factors to impact several aspects of GSCs. Our previous study demonstrated that M2 TAMs secrete PTN to promote self-renewal and survival of GSCs in GBMs[21]. These studies confirm that the molecular and cellular interactions between GSCs and TAMs are bi-directional.

Although CSF-1R inhibition has been shown to inhibits GBM tumor growth in a preclinical study[27], clinical trials using a CSF-1R inhibitor for cancer treatment failed due to its toxicity[66–68], because CSF-1R is expressed by many types of immune cells including monocytes[69,70]. Our study indicates that disruption of M2 TAM survival by targeting WISP1-related signaling may effectively suppress GBM tumor growth. As WISP1 is preferentially secreted by GSCs and maintains M2 TAM survival, and silencing WISP1 promotes apoptosis of M2 TAMs, targeting this GSC-specific paracrine signaling pathway to disrupt M2 TAMs may offer a therapeutic strategy to improve GBM

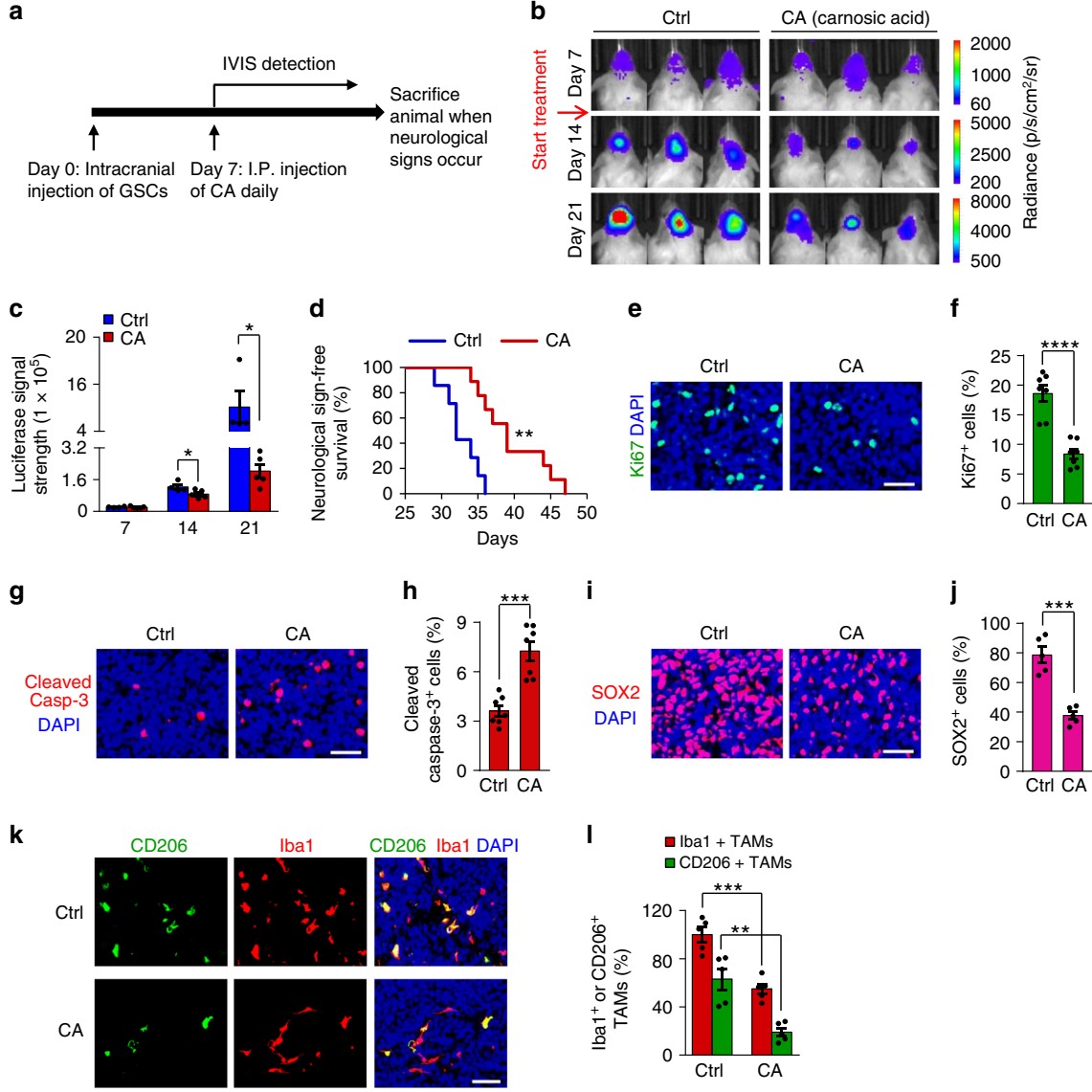

**Fig. 8 CA treatment reduced GSCs and M2 TAMs and inhibited GBM tumor growth. a** A schematic diagram showing the CA treatment of mice bearing the GBM xenografts derived from luciferase-expressing GSCs. **b, c** In vivo bioluminescent images (**b**) and quantification (**c**) of the T4121 GSC-derived xenografts after treatment with CA or the vehicle control at the indicated days after implantation. $n = 4$ (Ctrl) or 5 (CA) mice. Data are represented as mean ± s.e.m. *$p = 0.0482$ (day 14), *$p = 0.039$ (day 21), two-tailed unpaired $t$-test. p, photons; sr, steradian. **d** Kaplan–Meier survival curves of mice bearing T4121 GSC-derived xenografts treated with CA or the vehicle control. $n = 7$ (Ctrl) or 9 (CA) mice. CA-treated group vs. control group: **$p = 0.0011$, log-rank test. **e, f,** Immunofluorescent staining of Ki67 (green, **e**) and quantification of Ki67$^+$ cells (**f**) in T4121 GSC-derived tumors after treatment with CA or the vehicle control. $n = 7$ biological independent tumor tissues. Scale bar: 40 μm. Data are shown as mean ± s.e.m. ****$p < 0.0001$, two-tailed unpaired $t$-test. **g, h** Immunofluorescent staining of Cleaved Caspase-3 (red, **g**) and quantification of Cleaved Caspase-3$^+$ cells (**h**) in T4121 GSC-derived tumors after treatment with CA or the vehicle control. $n = 7$ biological independent tumor tissues. Scale bar: 40 μm. Data are represented as means ± s.e.m. ***$p = 0.0002$, two-tailed unpaired $t$-test. **i, j** Immunofluorescent staining of SOX2 (red, **i**) and quantification of SOX2$^+$ cells (**j**) in T4121 GSC-derived tumors after treatment with CA or the vehicle control. $n = 5$ biological independent tumor tissues. Scale bar: 40 μm. Data are represented as means ± s.e.m. ***$p = 0.0002$, two-tailed unpaired $t$-test. **k** Immunofluorescence staining of the M2 TAM Marker CD206 (green) and the pan-macrophage marker Iba1 (red) in T4121 GSC-derived tumors after treatment with CA or the vehicle control. Scale Bar, 50 μM. **l** Quantitation of CD206$^+$ TAM density and Iba1$^+$ total TAM density in xenografts treated with CA or the vehicle control. $n = 5$ biological independent tumor tissues. Data are shown as means ± s.e.m. **$p = 0.0017$, ***$p = 0.0003$; two-tailed unpaired $t$-test. Source data are provided as a Source data file.

treatment. Because there is no available WISP1 inhibitor so far and the Wnt/β-catenin signaling is activated in GSCs, we targeted the WISP1 upstream signaling with the β-catenin inhibitor carnosic acid for GBM treatment. As Wnt/β-catenin signaling induces multiple downstream targets to promote tumor growth, the inhibition of GBM growth by carnosic acid may be a comprehensive result. Nevertheless, carnosic acid treatment reduces WISP1 expression in vitro and in vivo, suggesting that WISP1

inhibition at least partially contributes to the therapeutic effect of carnosic acid. Because WISP1 is also expressed in other tumors[38,39] and may play a similar role in maintaining tumor-supportive M2 TAMs, targeting WISP1-associated signaling may improve treatment for other malignant tumors as well.

In summary, our study defined WISP1 as a key regulator in mediating the molecular crosstalk between GSCs and tumor-supportive M2 TAMs in the tumor microenvironment in GBMs.

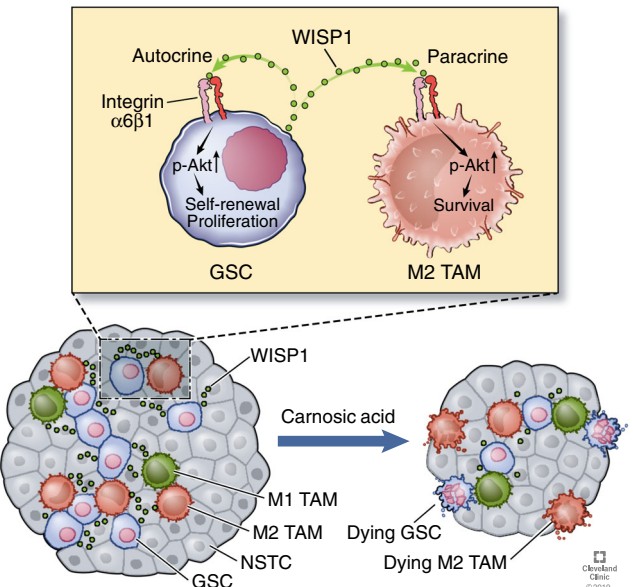

**Fig. 9 A schematic model showing autocrine and paracrine roles of WISP1 in GBM.** WISP1 is preferentially expressed and secreted by GSCs and promotes the GSC maintenance through an autocrine loop to activate Integrin α6β1-Akt signaling. GSC-secreted WISP1 also promotes M2 TAM survival through Integrin α6β1-mediated Akt activation in a paracrine manner. The dual role of WISP1 augments GBM malignant growth and progression. Targeting Wnt/β-catenin-WISP1 signaling with carnosic acid (CA) potently inhibits GBM tumor growth and may have therapeutic potential. Reprinted with permission, Cleveland Clinic Center for Medical Art & Photography © 2019–2020. All Rights Reserved.

We demonstrated that WISP1 plays both autocrine and paracrine roles in the maintenance of GSCs and in the survival of M2 TAMs, to promote malignant growth in GBMs. Disrupting WISP1 signaling or targeting its upstream regulators could potently suppress GBM growth through inhibition on both GSCs and tumor-supportive M2 TAMs, which may provide an effective therapeutic approach to improve treatment for GBMs and potentially other malignant tumors. In addition, as WISP1 is a secretory protein highly expressed by GBM tumors, WISP1 in serum or cerebrospinal fluid may serve as a promising diagnostic biomarker for GBMs or other cancers.

## Methods

**Human GBM specimens and glioma stem cells (GSCs).** Human primary GBM specimens in this study were collected from the Brain Tumor and Neuro-Oncology Center at Cleveland Clinic and University Hospitals of Case Western Reserve University in accordance with the Institutional Review Board-approved protocol. All procedures performed using human tissues were approved by the ethics committee of Cleveland Clinic and University Hospitals. Informed consent was obtained from individuals. GSCs and matched NSTCs were isolated from primary GBM specimens or patient-derived GBM xenografts and functionally characterized. Briefly, tumor cells were isolated from GBM tumors using Papain Dissociation System (Worthing Biochemical) according to the manufacturer's instructions and then were recovered in Neurobasal-A medium (Gibco) with B27 supplement (Gibco), 10 ng/ml EGF (Gold Biotech), 10 ng/ml bFGF (R&D), 1 mM sodium pyruvate (Gibco), and 2 mM L-glutamine (Gibco) at least 6 h. Isolated cells were labeled with a PE-conjugated anti-CD133 antibody (Miltenyi Biotec, 130-098-826) and a FITC-conjugated anti-CD15 antibody (BD, 347423) followed by FACs to sort the GSCs (CD15+/CD133+) or NSTCs (CD15-/CD133-). The cancer stem cell characteristics of isolated GSCs were validated by the expressions of GSC markers (SOX2, OLIG2, CD133, L1CAM) and a series of functional assays including serial neurosphere formation assay (in vitro limiting dilution assay), serum-induced cell differentiation assay and in vivo tumor formation limiting dilution assay. All experiments conform to relevant regulatory standards. Specifically, T387 GSCs and NSTCs were derived from a GBM from a 76-year old female patient. D456 GSCs and NSTCs were derived from a GBM from an 8-year old female patient. T4121 GSCs and NSTCs were derived from a GBM from a 53-year old male patient.

T3094 GSCs and NSTCs were derived from a GBM from a 63-year old male patient. T3565 GSCs and NSTCs were derived from a GBM from a 32-year old male patient. T3359 GSCs and NSTCs were derived from a GBM from a 31-year old male patient. CW1797 GBM specimens were collected from a 57-year old male patient. CW1798 GBM specimens were collected from a 47-year old male patient. CW2360 GBM specimens were collected from a 26-year old male patient. DI-74 GBM specimens were collected from a 52-year old male patient. CCF2445 GBM specimens were collected from a 50-year old male patient.

**Cell differentiation and in vivo limiting dilution assays.** For cell differentiation assay, GSCs were cultured on the Matrigel-coated dishes and induced for differentiation through withdrawal of all growth factors and by addition of serum (10% FBS in DMEM). At day 0, 2, 4, 6, 8, cells were harvested for immunoblot analysis or fixed for immunofluorescent staining of the GSC (SOX2, OLIG2) and differentiation markers (GFAP, MAP2). For in vivo limiting dilution assay, GSCs were counted and certain number cells (100, 500, 1000, 5000 or 10000) were implanted into the right frontal lobes of NSG mice. Mice were maintained up to 25 weeks or until the development of neurological signs. Brains of mice were collected, fixed in 4% paraformaldehyde, and paraffin embedded for hematoxylin-eosin staining.

**Intracranial tumorigenesis and treatment.** All animal procedures were approved by the Institutional Animal Care and Use Committee (IACUC) at Cleveland Clinic and were conducted in accordance with IACUC guidelines. Mice used in these studies were 6–7 weeks old female or male mice. NSG mice (The Jackson Laboratory) were housed under a 12-h light/12-h dark cycle in a temperature (20–26 °C) and humidity (30––70%) controlled environment and were fed ad libitum. 5000 GSCs were transplanted into the right cerebral cortex of NSG mice at a depth of 3.5 mm. Mice were monitored by the bioluminescent imaging or maintained until neurological signs were observed. For inducible overexpression, 5000 GSCs were transplanted intracranially into NSG mice for 10 days. The mice were then supplied with drinking water containing 2 mg/ml doxycycline or control water for 6 days. For the carnosic acid (Enzo Life Tech) treatment, 50 μL of 10 mg/kg carnosic acid was dissolved in DMSO and was administrated daily via intraperitoneal injection.

**Cell viability and tumorsphere formation assays.** For cell viability assay, 1000 cells were plated into each well of the 96-well plate, cell viability were determined at the indicated days after cell seeding using the Cell Titer-Glo Luminescent Cell Viability Assay kit (Promega) according to the manufacturer's protocol. For tumorsphere formation assay, 1000 GSCs were plated into each well of the 96-well plate, tumorsphere number was calculated at the sixth day after cell seeding.

**In vitro limiting dilution assays.** GSCs were plated into one well of 96-well plates at an indicated density (0, 4, 8, 12, 16 cells) with 30 replicates for each concentration. Six days later, the presence and number of tumorspheres in each well were recorded and analyzed using the software at http://bioinf.wehi.edu.au/software/elda/.

**Immunoblot analysis and phospho-kinase array.** For immunoblot analysis, we directly lysed cells or homogenised tissues in RIPA lysis buffer (50 mM Tris-HCl pH 7.4, 150 mM NaCl, 2 mM EDTA, 1% NP-40, 0.1% SDS and supplemented with protease inhibitors). Lystes were centrifuged for 10 min at 16,900 × g and 4 °C. The resulting supernatant fraction was separated by SDS-PAGE and transferred onto PVDF membranes. The membranes were blocked with 5% non-fat milk for 1 h and then immunoblotted with relative antibodies overnight at 4 °C followed by the HRP-conjugated antibody at room temperature for 1 h. Blots were imaged using BioRad Image Lab software. Phospho-kinase array was determined using the Proteome Profiler Human Phospho-Kinase Array Kit (R&D Systems). Briefly, cells were lysed in RIPA lysis buffer. Lystes were centrifuged for 10 min at 16,900 × g and 4 °C. Further analysis was performed according to the manufacturer's protocol. A complete list of antibodies including dilutions is shown in Supplementary Table 1. Uncropped images are shown in Supplementary Fig. 14.

**Co-immunoprecipitation (CoIP).** Cells were collected in IP lysis buffer (50 mM Tris-HCl pH 7.8, 137 mM NaCl, 1 mM EDTA, 1% Triton X-100, 10% glycerol and supplemented with protease inhibitors) for 30 min and pre-cleared by centrifugation at 16,900 × g for 10 min. Protein lysates were incubated with primary antibody or isotype IgG overnight at 4 °C and then captured by protein A/G Plus agarose beads (Santa cruz, sc-2003) for 2 h at 4 °C. The precipitants were washed with wash buffer (20 mM Tris-HCl pH 8.0, 0.2 mM EDTA, 100 mM KCl, 2 mM MgCl₂, 0.1% Tween 20, 10% glycerol) for 4 times, boiled with SDS sample buffer (50 mM Tris-HCL pH = 6.8, 2% SDS, 10% glycerol, 1% β-mercapitalethanol, 0.1% bromophenol blue) at 95 °C for ten minutes and subjected to immunoblot analysis. A complete list of antibodies including dilutions is shown in Supplementary Table 1. Uncropped images are shown in Supplementary Fig. 14.

**Conditioned medium preparation.** GSCs and matched NSTCs were cultured in neurobasal media without supplements and growth factors for 40 h. Conditioned medium was collected from cultures at a density of $2 \times 10^6$ cells/mL. The cells were

removed by centrifugation (300 × g, 5 min), and the conditioned medium was sterile filtered through a 0.2 μm filter. Samples were then concentrated to dryness by vacuum centrifugation using Eppendorf Concentrator plus/Vacufuge plus system (Eppendorf). Resulting residues were then dissolved in SDS sample buffer, denatured at 95 °C for ten minutes and then subjected to immunoblot analysis.

**DNA constructs and lentiviral transduction**. Lentiviral clones expressing two non-overlapping shRNAs against human WISP1 (TRCN0000373969, TRCN0000373970), human Integrin α6 (TRCN0000296162, TRCN000057775) and non-targeting shRNA (SHC002) were obtained from Sigma-Aldrich. A lentiviral construct expressing WISP1 was generated by cloning the human WISP1 open reading frame into the PCDH-MCs-T2A-Puro-MSCV vector (System Biosciences, CD522A-1) or PCW57.1 (Addgene, 50661). A lentiviral construct expressing myr-Akt1 was generated by cloning Akt1 with an N-terminal src myristoyation sequence into the PCDH-MCs-T2A-Puro-MSCV vector. Viral particles were produced in 293FT cells with pPACK set of helper plasmids (System Biosciences) in Neurobasal-A medium. The viruses were then concentrated by precipitation with PEG8000 (Fisher Scientific) according to the manufacturer's instructions. For lentiviral transduction, GSCs were transduced with lentivirus expressing the shRNA, WISP1 or Akt for 48 h, and then processed for next analysis.

**In vivo bioluminescence analysis**. To monitor tumor growth in living mice, GSCs were transduced with firefly luciferase through lentiviral infection. 48 h after shRNA infection, 5000 GSCs were intracranially transplanted into NSG mice. Then, mice were intraperitoneal injected with 120 mg/kg D-luciferin (Gold Biotech) and anesthetized with isoflurane at the indicated days. The size of the tumor was monitored by bioluminescence channel of IVIS Spectrum imaging system.

**Immunofluorescent staining**. Immunofluorescent staining were performed in tissues and cultured cells. Mouse GBM xenografts were collected from mice when neurological signs occur after GSC transplantation. Human GBM specimens were obtained from GBM patients through surgical resection. Briefly, clutured cells or tumor sectons were fixed in 4% PFA for 15 min and washed with PBS twice after that. Samples were blocked with a PBS solution containing 1% BSA plus 0.3% Triton X-100 for 30 minutes at room temperature, and then incubated with indicated primary antibody onvernight at 4 °C followed by the fluorescent second antibody (1:200) at room temperature for 2 h. Nuclei were counterstained with DAPI for 5 min, and then sections were mounted on glass and subjected to microscopy. Image J 1.47v (NIH) was used to quantify the positive cells. A complete list of antibodies including dilutions is shown in Supplementary Table 1.

**U937 cells and U937-derived M1 or M2 macrophages**. U937 cells were maintained in RPMI 1640 medium containing 10% Fetal Bovine Serum (FBS) at 37 °C in a humidified atmosphere with 5% $CO_2$. U937-derived M1 or M2 macrophages were generated as a macrophage model. Briefly, U937 cells were primed with PMA (Sigma, 5 nM) for 48 h to become unpolarized macrophages. To establish the M1 macrophages, the unpolarized macrophages were stimulated with 20 ng/ml of IFN-γ (Peprotech) and 100 ng/ml of LPS (Sigma) for an additional 48 h. To establish the M2 macrophages, the unpolarized macrophages were stimulated with IL4, IL10, and TGF- β (20 ng/ml, Peprotech) for additional 72 h. Cells were then harvested for immunoblot analysis or fixed for immunofluorescent staining of the indicated markers.

**Statistics and reproducibility**. Statistical differences were determined by two-tailed unpaired Student's t-test for two groups, or by two-way ANOVA for multiple groups. The data used in this study are presented as the mean ± s.d. or mean ± s.e.m. For Kaplan–Meier survival curves, statistical differences were determined by log-rank test. All analysis were carried out using Microsoft excel 2010 or GraphPad Prism 7 software. $p < 0.05$ was considered statistically significant. Detailed information is described in each figure legends. Except for the results from the public database, similar results were obtained from three independent experiments for all other results.

**Reporting summary**. Further information on research design is available in the Nature Research Reporting Summary linked to this article.

## Data availability

The TCGA database (Agilent-4502A platform) and Gravendeel database can be downloaded from GlioVis data portal (http://gliovis.bioinfo.cnio.es/). The gene expression in cluster of two-dimensional representation of cellular states can be downloaded from Single Cell Portal (http://singlecell.broadinstitute.org/single_cell/study/SCP393/single-cell-rna-seq-of-adult-and-pediatric-glioblastoma/). All other data supporting the findings of this study are available within the article and its Supplementary information files. All remaining data are available from the corresponding author upon reasonable request. The source data underlying Fig.1a–d, i, 2b, d–f, h, i, 3b, c, e, g, i, 4f, g, i, j, 5c–h, j, k, 7b, c, e, f, h, i, k, l, 8c, d, f, h, j, l, and Supplementary Figs. 1a, b, 2a, b, d–f, 3b, c, e, g, i, 4e, f, h, 5a–f, i, j, 6b, d, e, g, 7c, d, f, g,

8b, c, e, f, h, i, k, l, 9b, d, f, 10d, f, h, 12b, e, f, 13c–f, h, i, are provided as a Source data file. Source data are provided with this paper.

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

## Acknowledgements

We thank the Brain Tumor and Neuro-Oncology Centers at Cleveland Clinic and University Hospitals of Case Western Reserve University for providing GBM specimens for this study. We also thank the Flow Cytometry Core, Imaging Core, and Central Cell Services Core at Cleveland Clinic Lerner Research Institute for their assistance. This work was supported by Cleveland Clinic Foundation and the NIH R01 grants (NS091080 and NS099175) to S.B. and the NIH Shared Instrument Grant (S10OD018205) to the Cleveland Clinic Lerner Research Institute.

## Author contributions

W.T. and S.B. designed the experiments. W.T., C.C., W.Z., Z.H., K.Z., X.F., Q.H., A.Z., X.W., X.Y., H.H., and Q.W. performed the experiments. J.S.Y., X.L., G.R.S., and J.N.R. provided scientific input. A.E.S. provided some GBM surgical specimens. G.R.S. edited the paper. W.T. and S.B. analyzed the data and wrote the paper.

## Competing interests

The authors declare no competing interests.
