## [Peer Review File · Nature Communications]

Reviewers' comments:

Reviewer #1 (Remarks to the Author); expert in glioblastoma and mouse models:

In this manuscript, the authors, Tao, W. et al., report that a Wnt signaling-induced secreted protein, WISP1 promotes GBM tumorigenicity by stimulating functions of glioma stem cell and tumor-associated macrophages (TAMs). The authors first searched and identified WISP1 as Wnt-induced secreted protein from GSC but not non-GSC that displayed tumor-stimulating function on GSCs' tumorigenic behaviors and stemness in vitro and in vivo. The authors also identified alpha 6 beta 1 integrin as WISP1 receptor that mediated WISP1-stimulated p-Akt signaling. Then the authors turned to TAMs and identified and confirmed that WISP1- alpha 6 beta 1 integrin-Akt signaling also was responsible for GSC-stimulated recruitment and survival of TAMs into GSC tumor microenvironment. Lastly, targeting Wnt-induced WISP signaling using a Wnt inhibitor disrupted GSC stemness and M2 TAM survival and GSC tumorigenicity in vivo. This is a very interesting study with high significance. The authors identified and characterized a Wnt-induced secreted protein ligand derived from GSCs that acts upon GSC and TAMs for promoting their tumorigenic behaviors through GSC- and TMA-specific alpha 6 beta 1 integrin-Akt signaling. The methods and approaches as well as the techniques employed in this study are excellent. The data presentation is of high quality and strongly supportive to the conclusion. However, there are several weaknesses as described below. If the authors are able to address these comments, this manuscript could be further strengthened. The current enthusiasm for this study to be considered for publication in Nature Communications is moderate.

Major Comments:

1. There are necessary controls missing in several sets of experiments: 1) In Figure 4, a rescue experiment, i.e. shRNA KD of WISP1 then re-express shRNA resistant WISP1; 2) In Figure 5, treatment of GSCs with recombinant WISP1 in WISP1 KD GSCs; 3) effects of inhibition of alpha 6 and beta 1 by neutralizing antibodies on p-Akt;
2. Although the rationale of focusing on alpha 6 beta 1 integrin was based on previous reports, the controls of blocking other possible integrins, in particular, possible partners of beta subunit that known associate with alpha 6 integrin should be included.
3. The signaling identified here is Wnt-induced WISP1 that secreted out of GSC the acts upon GSC- and TMA-expressed alpha 6 beta 1 integrin-mediated p-Akt to promote GSC tumorigenicity. However, in Figures 7 and 8, the authors turned to target Wnt pathway for access this newly identified signaling in GSC and TMAs. Wnt is known to induce a plethora downstream targets. Among which, many are tumor-promoting. Thus the rational of targeting Wnt-pathway using the small molecule carnosic acid for this study is not strong. In addition, rational of selecting carnosic acid over other known Wnt inhibitors is also weak. Lastly whether the GSC model used in this study have enriched Wnt signaling is also not interrogated.

Reviewer #2 (Remarks to the Author); expert in Wnt and glioblastoma:

Dual Role of WISP1 in Maintaining Glioma Stem Cells and M2 Tumor-associated Macrophages to Promote Malignant Growth of Glioblastoma

This manuscript shows the role of secrete Wnt-induced signaling protein 1 (WISP1) in Glioma Stem Cells (GSCs) to facilitate a pro-tumor microenvironment by promoting the survival of both GSCs and tumor-associated macrophages (TAMs). Further, the important feature of this manuscript is that demonstrate that WISP1 signals through Integrin $\alpha 6\beta 1$ -Akt, in an autocrine fashion for GSCs and in a paracrine manner by M2 TAMs. The study is well-articulated and performed systematically. The major comments about this manuscript are as follows:

Comments

1. Figure 1 (f) which GSC culture was used in the experiment is not elaborated?
2. In Figure 1 (g), the method for preparation of conditioned medium is not elaborated. Was it a TCA precipitation method or just the conditioned medium? Please include molecular weight for all the immune-blot experiments.
3. In Figure 1 (e), Since these GSCs were maintaining stemness through Wnt/B-catenin active signaling and WISP1 is a downstream target gene, please include B-catenin expression in the immunoblot.
4. The authors state in that the shWISP1 cells did not develop tumors at all, as there was no sign of bioluminescence. It is not clear then as to why the shWISP1 cells injected mice were dying within 60-70 days after injections as shown in figure 3(c)?.
5. Figure 4h, mention the day of imaging.
6. Figure 5a and 5b, blocking with integrin antibodies may also be an indirect effect of integrin $\alpha 6 \beta 1$ receptor function in relation to WISP1 and is not sufficient to make a claim that WISP1 is acting as a ligand for $\alpha 6 \beta 1$ receptor. To confirm that WISP1 is acting as a ligand for integrin $\alpha 6 \beta 1$ receptor, it may be important to perform interaction studies to show that WISP1 associates with integrin $\alpha 6 \beta 1$ receptor. Also it is necessary to show the specificity of interaction of WISP1 with the receptor using a rescue experiment by using both recombinant WISP1 protein along with antibody at different ratios for the phosphorylation of AKT (Ser473).
7. In the experiments Figure 3a and 4h the authors have shown that there was no tumor development with ShWISP1 -1 and ShWISP1 -2 cells (no bioluminescence). It is then not clear as to how the authors have obtained xenograft sections that are used in Figures 6a to 6f?
8. The images are not arranged sequentially.
9. The manuscript talks of role of WISP1 in activation of Akt signaling in GSCs to promote cell proliferation and survival, which may partially augment tumor growth in vivo. This is not a novel finding as role of WISP1 in activation of Akt signaling is already known in several other cancers. Lu et al in 2016 in Eur J Pharmacol. 2016 Oct 5;788:90-97 has already shown that Akt signaling pathway mediates WISP1-induced migration and proliferation of human vascular smooth muscle cells. Another paper by Lukjanenko et al in Cell Stem Cell. 2019 Mar 7;24(3):433-446.e7 also recently showed that WISP1 is required for efficient muscle regeneration and controls the expansion and asymmetric commitment of muscle stem cells through Akt signaling. Thus, the role of WISP1 as activator of Akt signaling though not shown in glioma stem cells is already documented for other cell-types. Hence, this manuscript though shows rigor is not novel and hence may not be suitable for consideration for publication in this journal.

Reviewer #3 (Remarks to the Author); expert in macrophages and cancer:

Tao et al. describe interesting new results regarding the role of WISP1 in promoting glioblastoma progression. A clear and novel mechanistic framework is provided, arguing that WISP1 is specifically produced by glioma stem cells and provides an autocrine survival signal. In addition, WISP1 would also strongly promote the survival of M2 TAMs. The authors may want to consider the following comments.

1)

The authors use the TCGA and Gravendeel databases to examine the expression of WISP1 and other target genes in GBM. It would be interesting to also rely on the recently published human GBM single-cell RNAseq dataset (Neftel et al. 2019 Cell) to assess for the expression of WISP1 (and its putative integrin receptors) across the four GBM cellular states at single-cell resolution.

2) An important claim of the manuscript is that WISP1 signals through the Integrin $\alpha 6 \beta 1$ receptor. However, I feel that this needs to be substantiated:

a) The author could provide more direct evidence of WISP1- $\alpha 6 \beta 1$ interaction, for example via co-

immunoprecipitation experiments or more quantitatively via surface plasmon resonance or related techniques.

b) In addition, it is not clear why the integrin blocking studies were only performed in the WISP1 overexpression setting. Did the authors examine whether the addition of blocking antibodies inhibits GSC proliferation (without WISP1 overexpression), similar to what is seen in Fig 2b-d when silencing WISP1?

c) The authors could also use their sha6 construct to silence integrin $\alpha 6$ in GSCs (similar to what they did for U937 cells in Sup Fig 10). This should in theory phenocopy the WISP1 silencing of GSCs.

d) The authors report that silencing integrin $\alpha 6$ inhibits the proliferation of M2-polarized U937 cells. However, it is not clear to me where the WISP1 is coming from in this setting. Are the U937 cells producing WISP1 themselves or was this added to these cultures (which is not mentioned)? If the pro-survival effect in the U937 cells stems from WISP1- $\alpha 6\beta 1$ signaling, then why would just silencing $\alpha 6\beta 1$ in the absence of WISP1 lead to reduced survival?

3) While the results of WISP1 on GSCs are convincing and important. I am more hesitant with the proposed effects of WISP1 on TAMs. First, the reported effects of WISP1 silencing on TAMs seem quite dramatic, with a 60% reduction in total TAMs. It seems as if blocking WISP1 is more effective in obtaining GBM TAM depletion than CSF1R blockade. Indeed, it is reported that blocking CSF1R - one the most important macrophage growth factors - does actually not reduce the total number of TAMs in preclinical GBM (Pyonteck et al Nat Med 2013), showing that the tumor microenvironment can provide compensatory growth and survival signals. Here, the loss of WISP1, which is said to be specifically expressed in GSCs, seems sufficient to deplete the majority of TAMs throughout the tumor. The authors should at least try to speculate on the mechanism: which signaling pathways are disrupted?

When the density of TAMs is examined (Fig 6), it is not mentioned at which time point post GSC inoculation tumors were harvested. This is important since shWISP1 tumors grew much slower. Smaller tumors may have less (mature) TAMs, irrespective of paracrine WISP1 signaling. In the same line, WISP1 silencing may result in an altered tumor microenvironment (TME), which may attract less macrophages. Therefore, an alternative explanation for the lower macrophage density may be an altered TME (for example think of low vs. high grade tumors, where the latter contain significantly more TAMs), instead of a direct effect of WISP1 on TAM survival.

Second, the authors report that WISP1 very specifically augments survival of M2 TAMs, while it does not affect M1 TAMs. The macrophage field is increasingly realizing that the M1/M2 dichotomy in tumors (and other in vivo inflammation settings) is a major oversimplification. It needs to be taken into account that markers that are reported to adhere to M1 or M2 in one disease model may not necessarily do so in others (arguing for a spectrum model of macrophage activation, for example see Xue et al Immunity 2014). Additional complexity arises from the fact that the GBM TAM pool can exhibit a mixed ontogeny that partly dictates its transcriptional state and which again does not clearly adheres to M1/M2 (see Bowman et al 2016 Cell Reports, Chen et al 2017 Cancer Res.). Here, the authors use CD206 and CD11c as M2 and M1 markers, respectively. They report that around 60% of TAMs express CD206, while the additional 40% are CD11c+. To exemplify that relying on only a few markers can be problematic, consider mouse syngeneic GL261 GBM tumors, where the majority of TAMs are CD11c+, and a subset of CD11c+ TAMs co-express CD206 (for example see Peterson et al. 2016 PNAS). Therefore, in my opinion, in GL261 it would be problematic to just label CD11c+ TAMs as anti-tumoral M1. Of course, the xenografts reported in this manuscript may behave differently. In any case, to get a better understanding of the effect of WISP1 silencing on TAM heterogeneity in these xenografts, it would be very valuable if the authors were to perform a more in-depth analysis of the tumor myeloid cell pool, instead of relying on only a few markers in isolation. Multi-color flow cytometry can be very useful in this regard, especially when subsequently linked to an unbiased transcriptome analysis.

Minor comments

1) The authors may want to cite and discuss the work of Jing et al. *Int J. Onc* 2017, who already describe some of the tumor promoting roles of WISP1 in GBM.

Responses to Reviewer's Comments

We thank all reviewers for the critical evaluation of our manuscript. We greatly appreciate the insightful comments and helpful suggestions from the reviewers. In response to their major comments, we have performed a large amount of additional experiments and extensively revised the manuscript to address the critical issues. We believe that our manuscript has been significantly improved and strengthened now. Below, please find our point-by-point responses to the reviewers' comments. We hope that our responses have adequately addressed the important issues raised by the reviewers.

Reviewer #1 (Remarks to the Author); expert in glioblastoma and mouse models:

In this manuscript, the authors, Tao, W. et al., report that a Wnt signaling-induced secreted protein, WISP1 promotes GBM tumorigenicity by stimulating functions of glioma stem cell and tumor-associated macrophages (TAMs). The authors first searched and identified WISP1 as Wnt-induced secreted protein from GSC but not non-GSC that displayed tumor-stimulating function on GSCs' tumorigenic behaviors and stemness in vitro and in vivo. The authors also identified alpha 6 beta 1 integrin as WISP1 receptor that mediated WISP1-stimulated p-Akt signaling. Then the authors turned to TAMs and identified and confirmed that WISP1- alpha 6 beta 1 integrin-Akt signaling also was responsible for GSC-stimulated recruitment and survival of TAMs into GSC tumor microenvironment. Lastly, targeting Wnt-induced WISP signaling using a Wnt inhibitor disrupted GSC stemness and M2 TAM survival and GSC tumorigenicity in vivo. This is a very interesting study with high significance. The authors identified and characterized a Wnt-induced secreted protein ligand derived from GSCs that acts upon GSC and TAMs for promoting their tumorigenic behaviors through GSC- and TMA-specific alpha 6 beta 1 integrin-Akt signaling. The methods and approaches as well as the techniques employed in this study are excellent. The data presentation is of high quality and strongly supportive to the conclusion. However, there are several weaknesses as described below. If the authors are able to address these comments, this manuscript could be further strengthened. The current enthusiasm for this study to be considered for publication in Nature Communications is moderate.

Response: We are very grateful for the positive comments from the reviewer. Meanwhile, we greatly appreciate the helpful suggestions provided by the reviewer. We have performed additional experiments to address the important concerns.

Major Comments:

Reviewer #1: 1. There are necessary controls missing in several sets of experiments:

Reviewer #1: 1) In Figure 4, a rescue experiment, i.e. shRNA KD of WISP1 then re-express shRNA resistant WISP1;

Response: We thank the reviewer for the helpful suggestion and agree that necessary controls are critical. We used two independent shRNAs (shWISP1-1 and shWISP1-2) in this study. We checked sequences of these two shRNAs and found that shWISP1-2 targets the 3'-end non-coding region (3'-UTR) of *WISP1* mRNA to disrupt *WISP1* expression. As the *WISP1* overexpression construct does not contain the 3'-end non-coding sequence, we were able to simultaneously silence endogenous *WISP1* and overexpress exogenous *WISP1* in GSCs. Immunoblot analysis demonstrated that ectopic expression of *WISP1* (pCDH-*WISP1*) in GSCs rescued the decreased Akt phosphorylation (pAkt-Ser473) caused by knockdown of endogenous *WISP1* (Please see Figure R1 below).

Figure R1. Immunoblot analysis of Akt phosphorylation (pAkt-Ser473) in T4121 GSCs transduced with shWISP1-2 or shNT and then transduced with *WISP1* overexpression (pCDH-*WISP1*) or vector control. GSCs were transduced with shWISP1-2 or shNT for 36 hours and then transduced with *WISP1* overexpression or control vector for additional 36 hours through lentiviral infection.

We have added the new data in Supplementary Fig. 4b and described the result in our revised manuscript. Please see Line 204 at Page 9, the 4th part in the “Results” section: “As *WISP1* knockdown reduced Akt phosphorylation (Ser473), we further examined whether ectopic expression of *WISP1* rescues the effect induced by *WISP1* disruption.....Immunoblot analysis showed that ectopic expression of *WISP1* in GSCs rescued the decreased Akt phosphorylation (pAkt-Ser473) caused by knockdown of endogenous *WISP1* (Supplementary Fig. 4b).”

Reviewer #1: 2) In Figure 5, treatment of GSCs with recombinant WISP1 in WISP1 KD GSCs;

Response: We appreciate the insightful suggestion. We performed the suggested experiment and found that recombinant *WISP1* (r*WISP1*) treatment indeed rescued the decreased Akt

phosphorylation (p-Akt Ser473) caused by WISP1 disruption in a dose-dependent manner (Please see Figure R2 below).

We have added this additional data in Supplementary Fig. 4c and described the results in our revised manuscript. Please see Line 215 at Page 9, the 4th part in the “Results” section: “rWISP1 treatment also rescued the decreased Akt phosphorylation (pAkt-Ser473) caused by WISP1 disruption in a dose-dependent manner (Supplementary Fig. 4c).”

Reviewer #1: 3) effects of inhibition of alpha 6 and beta 1 by neutralizing antibodies on p-Akt;

Response: We thank the reviewer for the suggestion. We have performed the additional experiment and found that inhibiting Integrin $\alpha 6$ or $\beta 1$ by neutralizing antibody reduced Akt phosphorylation (Ser473) in GSCs (Please see Figure R3 below). However, inhibiting Integrin $\beta 4$, the other binding partner of Integrin $\alpha 6$, had no effect on Akt phosphorylation (pAkt-Ser473) in GSCs (Please see Figure R3 below).

We have added the new data in Supplementary Fig. 5g and described the results in our revised manuscript. Please see Line 255 at Page 11, the 5th part in the “Results” section: “Immunoblot analysis confirmed that inhibiting Integrin $\alpha 6$ or $\beta 1$ by blocking antibody reduced Akt phosphorylation (pAkt-Ser473) in GSCs, while inhibiting Integrin $\beta 4$ had no effect on Akt phosphorylation (pAkt-Ser473) (Supplementary Fig. 5g).”

Reviewer #1: 2. Although the rationale of focusing on alpha 6 beta 1 integrin was based on previous reports, the controls of blocking other possible integrins, in particular, possible partners of beta subunit that known associate with alpha 6 integrin should be included.

Response: We thank the reviewer for the important suggestion. Integrin $\alpha 6$ forms heterodimers with Integrin $\beta 1$ or $\beta 4$ ^{1, 2}. As shown in Figure R3 (above), inhibiting Integrin $\alpha 6$ or $\beta 1$ by its neutralizing antibody reduced Akt phosphorylation (pAkt-Ser473) in GSCs, but inhibiting Integrin $\beta 4$ has no effect on Akt phosphorylation (pAkt-Ser473). In addition, we examined the effects of inhibiting Integrin $\alpha 6$, $\beta 1$ or $\beta 4$ by blocking antibody on GSC proliferation and tumorsphere formation. The results showed that inhibiting Integrin $\alpha 6$ or $\beta 1$ significantly decreased GSC proliferation and tumorsphere formation, while inhibiting Integrin $\beta 4$ had no effect on GSC proliferation and tumorsphere formation (Please see Figure R4a, b below). Consistently, previous study has shown that Integrin $\beta 4$ is not expressed in GSCs². Thus, it is reasonable that inhibiting Integrin $\beta 4$ does not impact GSC proliferation.

We have added the new data in Fig. 5g, h and Supplementary Fig. 5e, f, and described the results in our revised manuscript. Please see Page 10, the 5th part in the “Results” section: “Moreover, treatment of GSCs with Integrin $\alpha 6$ or $\beta 1$ blocking antibody significantly decreased

GSC proliferation and tumorsphere formation (Fig. 5g, h and Supplementary Fig. 5e, f). However, blocking Integrin β 4, the other binding partner of Integrin α 6, had no effect on GSC proliferation and tumorsphere formation (Fig. 5g, h and Supplementary Fig. 5e, f)."

Reviewer #1: 3. The signaling identified here is Wnt-induced WISP1 that secreted out of GSC the acts upon GSC- and TMA-expressed alpha 6 beta 1 integrin-mediated p-Akt to promote GSC tumorigenicity. However, in Figures 7 and 8, the authors turned to target Wnt pathway for access this newly identified signaling in GSC and TMAs. Wnt is known to induce a plethora downstream targets. Among which, many are tumor-promoting. Thus the rational of targeting Wnt-pathway using the small molecule carnosic acid for this study is not strong. In addition, rational of selecting carnosic acid over other known Wnt inhibitors is also weak. Lastly whether the GSC model used in this study have enriched Wnt signaling is also not interrogated.

Response: We appreciate the critical concern. We have tried very hard to find the WISP1 inhibitor for this study, but there is no available WISP1 inhibitor so far. Thus, we had to target the Wnt/ β -catenin, the upstream signaling of WISP1. We understand that Wnt/ β -catenin signal may induce multiple downstream targets to promote tumor growth. Thus, the inhibition of GBM growth by carnosic acid may be a comprehensive result. Nevertheless, carnosic acid treatment reduced WISP1 expression *in vitro* and *in vivo*, suggesting that WISP1 inhibition may at least partially contribute to the therapeutic effect of carnosic acid. We have discussed it in the "Discussion" section. Please see Page 20, the 4th paragraph in the "Discussion" section: "Because there is no available WISP1 inhibitor so far and Wnt/ β -catenin signaling is activated in GSCs suggesting that WISP1 inhibition at least partially contribute the therapeutic effect of carnosic acid."

In addition, blood-brain barrier (BBB) prevents most anti-cancer agents from penetrating GBM tumors and limit therapeutic efficacy³. As carnosic acid can penetrate the blood brain barrier well^{4, 5}, and it has been reported to improve the treatment of medulloblastoma in mouse models⁵, we selected carnosic acid to test its effect of on GSCs, TAMs and GBM tumor growth in our models. We have described the reason of selecting carnosic acid for treatment in our revised manuscript. Please see Page 16, the last part in the "Results" section: "We selected the small molecule carnosic acid in our preclinical study, because it can penetrate the blood brain barrier and it has been reported to improve the treatment of medulloblastoma in a mouse model."

Furthermore, we performed immunoblot analysis and found that both total β -catenin and active β -catenin are enriched in all isolated GSC populations relative to matched non-stem tumor cells (NSTCs) (Please see Figure R5 below).

Figure R5. Immunoblot analyses of WISP1, Active β-catenin, β-catenin, SOX2 and OLIG2 expression in cell lysates of GSCs (+) and matched non-stem tumor cells (NSTCs) (-).

We have added the new data in Fig. 1e and described the result in our revised manuscript. Please see Page 6, the 1th part in the “Results” section: “Immunoblot analysis showed that WISP1, active β-catenin, total β-catenin and GSC markers including SOX2 and OLIG2 were preferentially expressed in GSCs relative to matched NSTCs (Fig. 1e).” This results provide us with the rationality of targeting Wnt/β-catenin signaling for GBM treatment. We have discussed it in the “Discussion” section. Please see Page 20, the 4th paragraph in the “Discussion” section: “Because there is no available inhibitor of WISP1 so far and Wnt/β-catenin signaling is activated in GSCssuggesting that WISP1 inhibition at least partially contribute to the therapeutic effect of carnosic acid.”

Reviewer #2 (Remarks to the Author); expert in Wnt and glioblastoma:

Dual Role of WISP1 in Maintaining Glioma Stem Cells and M2 Tumor-associated Macrophages to Promote Malignant Growth of Glioblastoma

This manuscript shows the role of secreted Wnt-induced signaling protein 1 (WISP1) in Glioma Stem Cells (GSCs) to facilitate a pro-tumor microenvironment by promoting the survival of both GSCs and tumor-associated macrophages (TAMs). Further, the important feature of this manuscript is that demonstrate that WISP1 signals through Integrin $\alpha6\beta1$ -Akt, in an autocrine fashion for GSCs and in a paracrine manner by M2 TAMs. The study is well-articulated and performed systematically. The major comments about this manuscript are as follows:

Response: We thank the reviewer for the positive comments. We appreciate the concerns and suggestions raised by the reviewer. We have performed additional experiments to address the reviewer's concerns.

Comments

Reviewer #2: 1. *Figure 1 (f) which GSC culture was used in the experiment is not elaborated?*

Response: We are sorry for missing the important information. We have added this information in Figure 1f.

Reviewer #2: 2. *In Figure 1 (g), the method for preparation of conditioned medium is not elaborated. Was it a TCA precipitation method or just the conditioned medium? Please include molecular weight for all the immune-blot experiments.*

Response: We regret that the method for the conditioned medium was not elaborated. We collected conditioned media from GSCs or NSTCs cultured in the Neurobasal medium without supplements and growth factors, and then concentrated conditioned media by using the Eppendorf Concentrator plus / Vacufuge vacuum centrifugation system. We have added the detailed description in the "Methods" section in the revised manuscript. Please see Line 591 at Page 24 in the "Conditioned Medium Preparation" section. In addition, we have added the molecular weight for all immunoblots.

Reviewer #2: 3. *In Figure 1 (e), Since these GSCs were maintaining stemness through Wnt/B-catenin active signaling and WISP1 is a downstream target gene, please include B-catenin expression in the immunoblot.*

Response: We thank the reviewer for the critical suggestion. We have performed immunoblot analysis and found that both total β -catenin and active β -catenin are enriched in all isolated GSCs relative to matched non-stem tumor cells (NSTCs) (Please see Figure R5 below).

Figure R5. Immunoblot analyses of WISP1, Active β-catenin, β-catenin, SOX2 and OLIG2 expression in cell lysates of GSCs (+) and matched non-stem tumor cells (NSTCs) (-).

We have added the new data in Fig. 1e and described the results in our revised manuscript. Please see Page 6, the 1st part in the “Results” section: “Immunoblot analysis showed that WISP1, active β-catenin, total β-catenin and GSC markers including SOX2 and OLIG2 were preferentially expressed in GSCs relative to matched NSTCs (Fig. 1e).”

Reviewer #2: 4. The authors state in that the shWISP1 cells did not develop tumors at all, as there was no sign of bioluminescence. It is not clear then as to why the shWISP1 cells injected mice were dying within 60-70 days after injections as shown in figure 3(c)?

Response: We thank the reviewer for raising this important issue. Because the luciferase signals from the brains of mice bearing the xenografts expressing shWISP1 were extremely weak within 30 days after transplantation, we could not detect obvious signals under bioluminescent imaging on day 14 and day 21 as shown in Figure 3a, although one of these mice in the shWISP1-1 group showed the luciferase signal at day 21. This did not mean that there were no micro-tumors in the brains of mice bearing xenografts expressing shWISP1. GSCs expressing shWISP1 proliferate slowly and need longer time to develop tumors in mouse brains. Indeed, the group of mice bearing xenografts expressing shWISP1 developed detectable tumors within 40-70 days after transplantation. Thus, the mice bearing xenografts expressing shWISP1 still died within 60-70 days although they took a relatively longer time (Figure 3c).

Reviewer #2: 5. Figure 4h, mention the day of imaging.

Response: We have added the day of imaging in Figure 4h.

Reviewer #2: 6. Figure 5a and 5b, blocking with integrin antibodies may also be an indirect effect

of integrin $\alpha6\beta1$ receptor function in relation to WISP1 and is not sufficient to make a claim that WISP1 is acting as a ligand for $\alpha6\beta1$ receptor. To confirm that WISP1 is acting as a ligand for integrin $\alpha6\beta1$ receptor, it may be important to perform interaction studies to show that WISP1 associates with integrin $\alpha6\beta1$ receptor. Also it is necessary to show the specificity of interaction of WISP1 with the receptor using a rescue experiment by using both recombinant WISP1 protein along with antibody at different ratios for the phosphorylation of AKT (Ser473).

Response: We thank the reviewer for the helpful suggestions.

(1) To validate that Integrin $\alpha6\beta1$ is a receptor for WISP1, we performed co-immunoprecipitation (CoIP) assay to confirm their binding as suggested by the reviewer. To increase the potential binding amounts, we overexpressed WISP1 in GSCs and then performed CoIP with anti-Integrin $\alpha6$ or $\beta1$ antibody. Anti-Integrin $\alpha6$ antibody pulled down the Integrin $\alpha6$ along with WISP1 and Integrin $\beta1$ (Please see Figure R6a below). In addition, the anti-Integrin $\beta1$ antibody also pulled down the Integrin $\beta1$ along with WISP1 and Integrin $\alpha6$ (Please see the Figure R6b below).

We have added the new data in Fig. 5l, m and Supplementary Fig. 5k, l and described the results in the revised manuscript. Please see Line 266 at Page 11, the 5th part in the “Results” section: “To validate that Integrin $\alpha 6\beta 1$ is a receptor for WISP1, we performed co-immunoprecipitation (CoIP) assay to confirm their binding.....In addition, the anti-Integrin $\beta 1$ antibody also pulled down the Integrin $\beta 1$ along with WISP1 and Integrin $\alpha 6$ (Fig. 5m and Supplementary Fig. 5l).”

(2) To test the specificity of the interaction between WISP1 and the receptor Integrin $\alpha 6\beta 1$, we examined Akt phosphorylation in GSCs treated with recombinant human WISP1 (rWISP1) protein along with Integrin blocking antibody at different ratios. Immunoblot analysis revealed that 5 $\mu\text{g/ml}$ Integrin $\alpha 6$ or $\beta 1$ blocking antibody dramatically prevented 0.2 $\mu\text{g/ml}$ rWISP1-induced Akt phosphorylation (pAkt-Ser473), while this dose of antibody had a relatively little effect on 0.8 $\mu\text{g/ml}$ rWISP1-induced Akt phosphorylation (pAkt-Ser473) (Please see Figure R7 below). However, 10 $\mu\text{g/ml}$ Integrin $\alpha 6$ or $\beta 1$ blocking antibody dramatically prevented increased Akt phosphorylation (pAkt-Ser473) induced by both doses of rWISP1 (Please see Figure R7 below). The results showed that Integrin $\alpha 6\beta 1$ is relatively specific to WISP1.

Figure R7. Immunoblot analysis of Akt phosphorylation (pAkt-Ser473) in T4121 GSCs treated with 5 or 10 $\mu\text{g/ml}$ Integrin blocking antibody in combination with 0.2 or 0.8 $\mu\text{g/ml}$ recombinant human WISP1 (rWISP1) protein. GSCs were cultured in the Neurobasal media without supplements for 12 hours, and then treated with treated with 5 or 10 $\mu\text{g/ml}$ Integrin blocking antibody in combination with 0.2 or 0.8 $\mu\text{g/ml}$ rWISP1 protein for 12 hours.

We have added this additional data in Supplementary Fig. 5m and described the results in our revised manuscript. Please see Line 272 at Page 11, the 5th part in the “Results” section: “To test the specificity of the interaction between WISP1 and the receptor Integrin $\alpha6\beta1$ The results showed that Integrin $\alpha6\beta1$ is relatively specific to WISP1.”

Reviewer #2: 7. In the experiments Figure 3a and 4h the authors have shown that there was no tumor development with ShWISP1 -1 and ShWISP1 -2 cells (no bioluminescence). It is then not clear as to how the authors have obtained xenograft sections that are used in Figures 6a to 6f?

Response: We appreciate the critical concern. We have addressed the similar issue in Comment #4. Since the luciferase signals from the brains of mice bearing the xenografts expressing shWISP1 were very weak within 30 days after transplantation, we could not detect obvious signals under bioluminescent imaging on day 14 and day 21 as shown in Figure 3a, although one of these mice in the shWISP1-1 group showed the luciferase signal at day 21. This did not indicate that there were no micro-tumors in the brains of mice bearing xenografts expressing shWISP1. GSCs expressing shWISP1 proliferate slowly and need longer time to develop tumors in mouse brains. Indeed, the group of mice bearing xenografts expressing shWISP1 developed detectable tumors within 40-70 days after transplantation. Thus, the mice bearing xenografts expressing shWISP1 still developed large tumors although they took a relatively longer time (Figure 3c). We collected the brains bearing GBM xenografts from mice when neurological signs occur. This time point is usually two to three days before the death of mice and the size of tumors

is large at this time point. Although we collected tumors from the control group (shNT) and shWISP1 groups at different times after transplantation, tumor sizes from these three groups (shNT, shWISP1-1 and shWISP1-2) were similar. Thus, these tumors from control and experimental groups were comparable for further analyses shown in Figure 6. We have described the collection time of tumor xenografts in the Legend part of Figure 6 (Page 43) and “Methods” section (Page 25) in our revised manuscript.

Reviewer #2: 8. The images are not arranged sequentially.

Response: We are sorry for that. We have tried our best to rearrange images or figure panels in sequential manner. Due to too many data panels in each figure, it is really hard to have all images or panels arranged sequentially in all figures. We sincerely hope that the reviewer understands the situation.

Reviewer #2: 9. The manuscript talks of role of WISP1 in activation of Akt signaling in GSCs to promote cell proliferation and survival, which may partially augment tumor growth in vivo. This is not a novel finding as role of WISP1 in activation of Akt signaling is already known in several other cancers. Lu et al in 2016 in Eur J Pharmacol. 2016 Oct 5;788:90-97 has already shown that Akt signaling pathway mediates WISP1-induced migration and proliferation of human vascular smooth muscle cells. Another paper by Lukjanenko et al in Cell Stem Cell. 2019 Mar 7;24(3):433-446.e7 also recently showed that WISP1 is required for efficient muscle regeneration and controls the expansion and asymmetric commitment of muscle stem cells through Akt signaling. Thus, the role of WISP1 as activator of Akt signaling though not shown in glioma stem cells is already documented for other cell-types. Hence, this manuscript though shows rigor is not novel and hence may not be suitable for consideration for publication in this journal.

Response: We appreciate the evaluation and helpful suggestions by the reviewer. We understand that the regulation of Akt signaling by WISP1 has been reported in other cell types. However, our study focused on the dual role of WISP1 in promoting both GSC and M2 TAM maintenance, thus supporting malignant growth of GBM. The role of WISP1 in maintaining the tumor-supportive TAMs (M2) to promote GBM tumor growth has not been reported. This is one of novel points of this manuscript. In addition, we first found that Integrin $\alpha 6\beta 1$ is the receptor of WISP1 in both GSCs and M2 TAMs in GBMs, and identified that WISP1- $\alpha 6\beta 1$ -Akt signaling is responsible for GSC-promoted survival of TAMs in tumor microenvironment, which is another new point. Furthermore, we found targeting the Wnt/ β -catenin-WISP1 signaling with carnosic acid potently inhibited GBM tumor growth and extended the survival of tumor-bearing mice, suggesting that targeting this signaling axis represents an attractive therapeutic strategy. Therefore, we believe that our findings in this manuscript are significant and contain novel points, which will make it interesting to general readers.

Reviewer #3 (Remarks to the Author); expert in macrophages and cancer:

Tao et al. describe interesting new results regarding the role of WISP1 in promoting glioblastoma progression. A clear and novel mechanistic framework is provided, arguing that WISP1 is specifically produced by glioma stem cells and provides an autocrine survival signal. In addition, WISP1 would also strongly promote the survival of M2 TAMs. The authors may want to consider the following comments.

Response: We thank the reviewer for the time and effort to review our manuscript. We appreciate the concerns and suggestions raised by the reviewer. We have performed a large amount of additional experiments to address the reviewer's concerns. We believe that this manuscript is significantly improved after extensive revision in response to the constructive suggestions.

Reviewer #3: 1) *The authors use the TCGA and Gravendeel databases to examine the expression of WISP1 and other target genes in GBM. It would be interesting to also rely on the recently published human GBM single-cell RNAseq dataset (Neffel et al. 2019 Cell) to assess for the expression of WISP1 (and its putative integrin receptors) across the four GBM cellular states at single-cell resolution.*

Response: We thank the reviewer for the important suggestion. We examined the expression of WISP1 and Integrin $\alpha 6\beta 1$ across the four GBM cellular states. The results showed that WISP1 is enriched in some AC-like and MES-like cells, while Integrin $\alpha 6$ and $\beta 1$ are widely expressed in all four states (Please see Figure R8 below). These data suggest that WISP1 and Integrin $\alpha 6\beta 1$ are at least co-expressed by some AC-like and MES-like cells in GBM.

Figure R8. The expression of WISP1, Integrin $\alpha 6$ and $\beta 1$ in cluster of two-dimensional representation of cellular states. Each quadrant corresponds to one cellular state, the exact

position of malignant cells (dots) reflect their relative scores for the meta-modules, and their colors reflect the gene expression levels.

We have added the new data in Supplementary Fig. 5n and described the results in our revised manuscript. Please see Line 283 at Page 12, the 5th part in the “Results” section: “A recent study has shown that malignant cells in human GBM exist in four main cellular states that recapitulate neural-progenitor-like (NPC-like), oligodendrocyte-progenitor-like (OPC-like), astrocyte-like (AC-like), and mesenchymal-like (MES-like) states.....These data suggest that WISP1 and Integrin $\alpha6\beta1$ are co-expressed by some AC-like and MES-like cells in GBM (Supplementary Fig. 5n).”

Reviewer #3: 2) An important claim of the manuscript is that WISP1 signals through the Integrin $\alpha6\beta1$ receptor. However, I feel that this needs to be substantiated:

a) The author could provide more direct evidence of WISP1- $\alpha6\beta1$ interaction, for example via co-immunoprecipitation experiments or more quantitatively via surface plasmon resonance or related techniques.

Response: We thank the reviewer for the insightful suggestion. To confirm that Integrin $\alpha6\beta1$ is a receptor for WISP1, we performed co-immunoprecipitation (CoIP) assay to confirm their binding. To increase the potential binding amounts, we overexpressed WISP1 in GSCs and then performed CoIP with anti-Integrin $\alpha6$ or $\beta1$ antibody. The anti-Integrin $\alpha6$ antibody pulled down the Integrin $\alpha6$ along with WISP1 and Integrin $\beta1$ (Please see Figure R6a below). In addition, the anti-Integrin $\beta1$ antibody also pulled down the Integrin $\beta1$ along with WISP1 and Integrin $\alpha6$ (Please see Figure R6b below).

Figure R6. a,b, CoIP assays of protein interaction in GSCs transduced with WISP1 for overexpression through lentiviral infection. Cell lysates were immunoprecipitated (IP) with anti-Integrin $\alpha 6$ (a) or anti-Integrin $\beta 1$ (b) antibody and then immunoblotted with anti-WISP1, anti-Integrin $\alpha 6$ and anti-Integrin $\beta 1$ antibodies.

We have added the new data in Fig. 5l, m and Supplementary Fig. 5k, l and described the results in our revised manuscript. Please see Line 266 at Page 11, the 5th part in the “Results” section: “To validate that Integrin $\alpha 6\beta 1$ is a receptor for WISP1, we performed co-immunoprecipitation (CoIP) assay to confirm their binding.....In addition, the anti-Integrin $\beta 1$ antibody also pulled down the Integrin $\beta 1$ along with WISP1 and Integrin $\alpha 6$ (Fig. 5m and Supplementary Fig. 5l).”

Reviewer #3: 2)-(b) In addition, it is not clear why the integrin blocking studies were only performed in the WISP1 overexpression setting. Did the authors examine whether the addition of blocking antibodies inhibits GSC proliferation (without WISP1 overexpression), similar to what is seen in Fig 2b-d when silencing WISP1?

Response: We thank the reviewer for raising this important point. Integrin $\alpha 6$ forms heterodimers with Integrin $\beta 1$ or $\beta 4$ ^{1,2}. We also used Integrin $\beta 4$ blocking antibody to perform the experiment following the suggestion from another reviewer. We examined the effects of inhibiting Integrin $\alpha 6$, $\beta 1$ or $\beta 4$ by blocking antibody on GSC proliferation and tumorsphere formation. The results showed that inhibiting Integrin $\alpha 6$ or $\beta 1$ significantly decreased GSC proliferation and tumorsphere formation, while inhibiting Integrin $\beta 4$ had no effect on GSC proliferation and tumorsphere formation (Please see Figure R4a, b below). Consistently, previous study has shown that Integrin $\beta 4$ is not expressed in GSCs². Thus, it is reasonable that inhibiting Integrin $\beta 4$ showed

no effect on GSC proliferation.

We have added the new data in Fig. 5g, h and Supplementary Fig. 5e, f and described the results in our revised manuscript. Please see Page 10, the 5th part in the “Results” section: “Moreover, treatment of GSCs with Integrin α6 or β1 blocking antibody significantly decreased GSC proliferation and tumorsphere formation (Fig. 5g, h and Supplementary Fig. 5e, f). However, blocking Integrin β4, the other binding partner of Integrin α6, had no effect on GSC proliferation and tumorsphere formation (Fig. 5g, h and Supplementary Fig. 5e, f).”

We also examined Akt phosphorylation (pAkt-Ser473) in GSCs treated with Integrin blocking antibody. The results showed that inhibiting Integrin α6 or β1 by blocking antibody reduced Akt phosphorylation (pAkt-Ser473) in GSCs, while inhibiting Integrin β4, the other binding partner of Integrin α6, had no effect on Akt phosphorylation (pAkt-Ser473) in GSCs (Please see Figure R3 below).

Integrin blocking antibody (5 μ g/ml) or isotype IgG control for 12 hours.

We have added the additional data in Supplementary Fig. 5g and described the results in our revised manuscript. Please see line 255 at Page 11, the 5th part in the “Results” section: “Immunoblot analysis confirmed that inhibiting Integrin α 6 or β 1 by blocking antibody reduced Akt phosphorylation (pAkt-Ser473) in GSCs, while inhibiting Integrin β 4 had no effect on Akt phosphorylation (pAkt-Ser473) (Supplementary Fig. 5g).”

Reviewer #3: 2)-(c) The authors could also use their sha6 construct to silence integrin α 6 in GSCs (similar to what they did for U937 cells in Sup Fig 10). This should in theory phenocopy the WISP1 silencing of GSCs.

Response: We appreciate the helpful suggestion. We examined the effects of Integrin α 6 disruption by shRNA on GSC proliferation and Akt phosphorylation. shRNAs targeting α 6 significantly decreased Integrin α 6 expression and Akt phosphorylation (pAkt-Ser473) in GSCs (Please see Figure R9a below). Disruption of Integrin α 6 also significantly inhibited GSC proliferation and tumorsphere formation (Please see Figure R9b, c below).

Figure R9. a, Immunoblot analysis of Akt phosphorylation (pAkt-Ser473) and Integrin α6 expression in GSCs transduced with shIntegrin α6 or shNT control.

b, Cell viability assay of GSCs transduced with shIntegrin α6 or shNT. Data are shown as means ± s.d. shIntegrin α6 versus shNT, *** $p < 0.001$, two-tailed unpaired t -test.

c, Tumorsphere formation of T4121 GSCs transduced with shIntegrin α6 or shNT. Data are shown as means ± s.d. *** $p < 0.001$

We have added the new data in Fig. 5i-k and Supplementary Fig. 5h-j and described the results in our revised manuscript. Please see Line 258 at Page 11, the 5th part in the “Results” section: “We next examined the effects of Integrin α6 disruption by shRNA on GSC proliferation and Akt phosphorylation.....Disruption of Integrin α6 also significantly inhibited GSC proliferation and tumorsphere formation (Fig. 5i-k and Supplementary Fig. 5h-j).”

Reviewer #3: 2)-(d) The authors report that silencing integrin α6 inhibits the proliferation of M2-polarized U937 cells. However, it is not clear to me where the WISP1 is coming from in this setting. Are the U937 cells producing WISP1 themselves or was this added to these cultures (which is not mentioned)? If the pro-survival effect in the U937 cells stems from WISP1-α6β1 signaling, then why would just silencing α6β1 in the absence of WISP1 lead to reduced survival?

Response: We thank the reviewer for the important question. We examined WISP1 expression in U937-derived M1 or M2 macrophages and found that WISP1 was not expressed in both U937-derived M1 and M2 macrophages (Please see Figure R10 below).

Figure R10. Immunoblot analysis of WISP1 expression in U937-derived M1 and M2 macrophages.

Therefore, we added human recombinant WISP1 (rWISP1) protein in this experimental setting and performed the experiment (Please see Figure R11 below).

Figure R11. Cell viability assay of U937-derived M2 macrophages transduced with shNT or shIntegrin $\alpha 6$ and cultured with rWISP1 protein. Cells were infected with shNT or shIntegrin $\alpha 6$ lentivirus for 24 hours and then cultured in serum-free media with rWISP1 protein (400 $\mu\text{g}/\text{ml}$) for 4 days. Data are represented as means \pm s.d. *** $p < 0.001$, two-tailed unpaired t -test.

We found that disruption of Integrin $\alpha 6$ by shRNAs inhibited the rWISP1-enhanced survival of M2 macrophages when cultured under serum starvation conditions (Figure R11).

We have added the new data in Supplementary Fig. 12e and described the results in our revised manuscript. Please see Page 15, the 7th part in the “Results” section: “To assess whether WISP1 promotes the survival of M2 TAMs through Integrin $\alpha 6\beta 1$ signaling..... Disruption of Integrin $\alpha 6$ by shRNAs inhibited the rWISP1-enhanced survival of M2 macrophages when cultured under serum starvation conditions (Supplementary Fig. 12e).”

Reviewer #3: 3) While the results of WISP1 on GSCs are convincing and important. I am more hesitant with the proposed effects of WISP1 on TAMs. First, the reported effects of WISP1 silencing on TAMs seem quite dramatic, with a 60% reduction in total TAMs. It seems as if blocking WISP1 is more effective in obtaining GBM TAM depletion than CSF1R blockade. Indeed, it is reported that blocking CSF1R - one the most important macrophage growth factors - does actually not reduce the total number of TAMs in preclinical GBM (Pyonteck et al Nat Med 2013),

showing that the tumor microenvironment can provide compensatory growth and survival signals. Here, the loss of WISP1, which is said to be specifically expressed in GSCs, seems sufficient to deplete the majority of TAMs throughout the tumor. The authors should at least try to speculate on the mechanism: which signaling pathways are disrupted? When the density of TAMs is examined (Fig 6), it is not mentioned at which time point post GSC inoculation tumors were harvested. This is important since shWISP1 tumors grew much slower. Smaller tumors may have less (mature) TAMs, irrespective of paracrine WISP1 signaling. In the same line, WISP1 silencing may result in an altered tumor microenvironment (TME), which may attract less macrophages. Therefore, an alternative explanation for the lower macrophage density may be an altered TME (for example think of low vs. high grade tumors, where the latter contain significantly more TAMs), instead of a direct effect of WISP1 on TAM survival.

Response: We thank the reviewer for raising the important point. We have proposed the molecular mechanisms underlying the WISP1-promoted maintenance and survival of M2 TAMs. Our *in vivo* and *in vitro* results demonstrate that WISP1 secreted by GSCs promoted TAM survival through the Integrin $\alpha6\beta1$ -Akt signaling. Our *in vivo* experiments validated that silencing WISP1 markedly reduced TAM density, particularly tumor-supportive TAMs.

We collected the brains bearing GBM xenografts from mice when neurological signs occur. This time point is usually two to three days before the death of mice and the size of tumors is large at this time point. Because tumor developed much slower in xenografts expressing shWISP1, we collected tumors from the control group (shNT) and shWISP1 groups at different times after transplantation. Thus, tumor sizes from these three groups (shNT, shWISP1-1 and shWISP1-2) were similar when they were harvested, which enables that those tumors from control and experimental groups were comparable for analyses of TAM density. We have described the collection time of tumor xenografts in the Legend part of Figure 6, Supplementary Figure 7c-h, Supplementary Figure 8 and “Methods” section (Page 25) in our revised manuscript.

The reviewer mentioned that silencing WISP1 may result in an altered tumor microenvironment, which may contribute to decreased TAM number. According to our data, we cannot rule out this possibility. However, our *in vitro* data suggest that WISP1 has a direct effect on the survival of macrophages. It would be interesting to further investigate whether WISP1 can regulate the tumor microenvironment in GBM in the future study. We have discussed these issues in the “Discussion” section. Please see Line 454 at Page 18, the 3rd paragraph in the “Discussion” section: “Previous study reported that blocking CSF1R..... It would be interesting to further investigate whether WISP1 can regulate the tumor microenvironment in GBMs in the future.”

Reviewer #3: 3). Second, the authors report that WISP1 very specifically augments survival of M2 TAMs, while it does not affect M1 TAMs. The macrophage field is increasingly realizing that the M1/M2 dichotomy in tumors (and other in vivo inflammation settings) is a major oversimplification.

It needs to be taken into account that markers that are reported to adhere to M1 or M2 in one disease model may not necessarily do so in others (arguing for a spectrum model of macrophage activation, for example see Xue et al Immunity 2014). Additional complexity arises from the fact that the GBM TAM pool can exhibit a mixed ontogeny that partly dictates its transcriptional state and which again does not clearly adheres to M1/M2 (see Bowman et al 2016 Cell Reports, Chen et al 2017 Cancer Res.). Here, the authors use CD206 and CD11c as M2 and M1 markers, respectively. They report that around 60% of TAMs express CD206, while the additional 40% are CD11c+. To exemplify that relying on only a few markers can be problematic, consider mouse syngeneic GL261 GBM tumors, where the majority of TAMs are CD11c+, and a subset of CD11c+ TAMs co-express CD206 (for example see Peterson et al. 2016 PNAS). Therefore, in my opinion, in GL261 it would be problematic to just label CD11c+ TAMs as anti-tumoral M1. Of course, the xenografts reported in this manuscript may behave differently. In any case, to get a better understanding of the effect of WISP1 silencing on TAM heterogeneity in these xenografts, it would be very valuable if the authors were to perform a more in-depth analysis of the tumor myeloid cell pool, instead of relying on only a few markers in isolation. Multi-color flow cytometry can be very useful in this regard, especially when subsequently linked to an unbiased transcriptome analysis.

Response: We appreciate the important concern raised by the reviewer. We agree with the reviewer that the M1/M2 dichotomy is an oversimplification of TAMs in tumors. We also think that there is a heterogeneity of TAMs in GBM tumors. In this study, we used the term “M2 TAMs” to indicate the tumor-supportive macrophages that may contain several subpopulations, and used “M1 TAMs” to indicate the tumor-suppressive macrophages that may also contain several subpopulations. The M1/M2 dichotomy used in this manuscript does not mean that there are only two types of TAMs in GBM tumors. We also agree that M1 or M2 markers in one disease model may not necessarily be the same in other disease models, as the previous work showing that the majority of TAMs are CD11c⁺, and a subset of CD11c⁺ TAMs co-express CD206⁶. In our study, we used CD206 and CD163 as M2 TAM markers, and CD11c and CD16/32 as M1 TAM markers. According to our previous results, CD206⁺ and CD163⁺ TAMs may represent one major sub-population of TAMs, and CD11c⁺ and CD16/32⁺ TAMs are the another major sub-population. To further verify this point, we performed immunofluorescent staining in GBM xenografts using these markers. We found that more than 90% CD206⁺ TAMs express CD163 (Please see Figure R12a, b below), and more than 90% CD11c⁺ TAMs express CD16/32 (Please see Figure R12c, d below). These data indicate that CD206⁺ and CD163⁺ TAMs are almost the same population, and CD11c⁺ and CD16/32⁺ TAMs are nearly the same population.

We next performed immunofluorescent staining in GBM xenografts using CD206 and CD11c antibodies and found that less than 6% CD206⁺ TAMs express CD11c (Please see Figure R13 below). This data further confirm that CD206⁺/CD163⁺ and CD11c⁺/CD16/32⁺ TAMs represent very different sub-populations of TAMs. Our previous work has demonstrated that CD163⁺ TAMs promote GBM tumor growth in our xenograft models (T4121 and T387 GSC-derived xenografts)^{7, 8}. Therefore, these studies further confirm that silencing WISP1 indeed reduced the number of tumor-supportive macrophages (M2 TAMs) in our xenograft models. According to these results, we think that M2/M1 TAMs indeed represent two major but functionally different macrophage populations (Tumor-supportive and tumor-suppressive macrophages) in our tumor models, although we can't rule out that each major population (M2 or M1) may contain several subpopulations.

To further confirm that disrupting WISP1 specifically decreased M2 tumor-supportive macrophages, we used additional two M2 markers (Arg1 and Fizz1) and two M1 markers (iNOS and MHCII). We selected these markers for the study because they have been used to distinguish M2/M1 TAMs in our GBM xenograft models⁸ and some other GBM xenograft models⁹⁻¹¹. We also found that about 60% TAMs express Arg1 or Fizz1, and that WISP1 disruption markedly reduced Arg1⁺ or Fizz1⁺ M2 TAMs in GSC-derived tumors (please see Figure R14a-f below). These results are consistent with our previous results using CD206 and CD163 as M2 markers (Fig. 6g-l). In addition, disrupting WISP1 had little effect on iNOS⁺ or MHCII⁺ M1 TAMs (please see Figure R15a-f below).

Taken together, all these data demonstrated that silencing WISP1 indeed decreased tumor-supportive M2 macrophages in GSC-derived xenografts, while had little effect on tumor-suppressive M1 macrophages. We believe that our results will provide some useful information

for therapeutic targeting of tumor-supportive macrophages (M2 TAMs) in GBMs. We have added these new data in Supplementary Fig. 7c-h and Supplementary Fig. 8g-l and revised the description of this results in our revised manuscript. Please see Line 320 at Page 13, the 6th part in the “Results” section: “We used specific M2 markers (CD206, CD163, Arg1 and Fizz1) and M1 markers (CD11c, CD16/32, iNOS and MHCII) for the study, as those markers have been used to distinguish M2/M1 TAMs in our GBM xenograft models and some other GBM xenograft models.” In addition, we added the following in the Discussion part at Page 18 (Line 441): “We fully recognized that the M1/M2 dichotomy is an oversimplification of TAMs in tumors. In this study, we just used the term “M2 TAMs” to indicate the tumor-supportive macrophages that may contain several subpopulations, and used “M1 TAMs” to represent the tumor-suppressive macrophages that may also contain subpopulations. The M1/M2 dichotomy used here does not mean that there are only two simple types of TAMs in GBM tumors. We believe that there is a heterogeneity of TAMs in GBM tumors. However, our studies confirmed that silencing WISP1 indeed reduced tumor-supportive macrophages (M2 TAMs) in our xenograft models. According to our previous studies and current data, it is reasonable to conclude that M2/M1 TAMs indeed represent two major but functionally different macrophage populations (Tumor-supportive and tumor-suppressive macrophages) in our tumor models, although we can’t rule out that each major population (M2 or M1) may contain several subpopulations. It will be interesting to further analyze subpopulations in M2 TAMs and M1 TAMs in GBMs in the future.”

Minor comments

Reviewer #3: 1) The authors may want to cite and discuss the work of Jing et al. Int J. Onc 2017, who already describe some of the tumor promoting roles of WISP1 in GBM.

Response: We are sorry for missing the important reference. We have discussed it in the “Discussion” section and cited it. Please see Page 17, the 2nd paragraph in the “Discussion” section: “Recent study has also demonstrated that WISP1 is a novel oncogene in GBM.....the origin of WISP1 in GBM and the role of WISP1 in regulating of GSC properties remain unclear.”

Reference:

1. Velozo, J. *et al.* Severe alterations in expression and localisation of $\alpha 6 \beta 4$ integrin in salivary gland acini from patients with Sjogren syndrome. *Annals of the rheumatic diseases* **68**, 991-996 (2009).
2. Lathia, J.D. *et al.* Integrin alpha 6 regulates glioblastoma stem cells. *Cell stem cell* **6**, 421-432 (2010).
3. Zhou, W. *et al.* Targeting Glioma Stem Cell-Derived Pericytes Disrupts the Blood-Tumor Barrier and Improves Chemotherapeutic Efficacy. *Cell stem cell* **21**, 591-603 e594 (2017).
4. Azad, N., Rasoolijazi, H., Joghataie, M.T. & Soleimani, S. Neuroprotective effects of carnosic Acid in an experimental model of Alzheimer's disease in rats. *Cell journal* **13**, 39-44 (2011).
5. Rodriguez-Blanco, J. *et al.* Inhibition of WNT signaling attenuates self-renewal of SHH-subgroup medulloblastoma. *Oncogene* **36**, 6306-6314 (2017).
6. Peterson, T.E. *et al.* Dual inhibition of Ang-2 and VEGF receptors normalizes tumor vasculature and prolongs survival in glioblastoma by altering macrophages. *Proceedings of the National Academy of Sciences of the United States of America* **113**, 4470-4475 (2016).
7. Shi, Y. *et al.* Tumour-associated macrophages secrete pleiotrophin to promote PTPRZ1 signalling in glioblastoma stem cells for tumour growth. *Nature communications* **8**, 15080 (2017).
8. Zhou, W. *et al.* Periostin secreted by glioblastoma stem cells recruits M2 tumour-associated macrophages and promotes malignant growth. *Nature cell biology* **17**, 170-182 (2015).
9. De, I. *et al.* CSF1 Overexpression Promotes High-Grade Glioma Formation without Impacting the Polarization Status of Glioma-Associated Microglia and Macrophages. *Cancer research* **76**, 2552-2560 (2016).
10. Grimaldi, A. *et al.* KCa3.1 inhibition switches the phenotype of glioma-infiltrating microglia/macrophages. *Cell death & disease* **7**, e2174 (2016).
11. Xue, N. *et al.* Chlorogenic acid inhibits glioblastoma growth through repolarizing macrophage from M2 to M1 phenotype. *Scientific reports* **7**, 39011 (2017).

REVIEWERS' COMMENTS:

Reviewer #1 (Remarks to the Author):

In the revised manuscript, the authors have satisfactorily addressed all the comments I raised with strong new data and/or reasonable explanation and corresponding changes in the result descriptions and/or discussions. This is a much improved study with further strengthened significance. This study is sufficient for its publication in Nature Communications.

Reviewer #2 (Remarks to the Author):

The authors have answered all the concerns raised by the reviewers. I am satisfied with the experiments done by the authors in support of the comments posed. The experimental data generated is satisfactory and clarifies the questions raised. I have no further queries regarding this manuscript and I recommend its acceptance by the journal Nature Communications.

Dr. Anjali Shiras;
Principal Investigator,
National Centre for Cell Science (NCCS),
Pune-411007,
INDIA.

Reviewer #3 (Remarks to the Author):

The authors have very nicely addressed all the comments that were raised. New data has been added that strengthens the manuscript. This is important and interesting work and I support acceptance of the manuscript and publication in Nature Communications.

Responses to Reviewers' Comments

REVIEWERS' COMMENTS:

Reviewer #1 (Remarks to the Author):

In the revised manuscript, the authors have satisfactorily addressed all the comments I raised with strong new data and/or reasonable explanation and corresponding changes in the result descriptions and/or discussions. This is a much improved study with further strengthened significance. This study is sufficient for its publication in Nature Communications.

Response: We are very grateful for the positive comments from the reviewer.

Reviewer #2 (Remarks to the Author):

The authors have answered all the concerns raised by the reviewers. I am satisfied with the experiments done by the authors in support of the comments posed. The experimental data generated is satisfactory and clarifies the questions raised. I have no further queries regarding this manuscript and I recommend its acceptance by the journal Nature Communications.

Response: We thank the reviewer for the positive comments.

Reviewer #3 (Remarks to the Author):

The authors have very nicely addressed all the comments that were raised. New data has been added that strengthens the manuscript. This is important and interesting work and I support acceptance of the manuscript and publication in Nature Communications.

Response: We appreciate the reviewer's positive comments.